**TOOLS**

# Probing the subcellular distribution of phosphatidylinositol reveals a surprising lack at the plasma membrane

James P. Zewe, April M. Miller, Sahana Sangappa, Rachel C. Wills, Brady D. Goulden, and Gerald R.V. Hammond

**The polyphosphoinositides (PPIn) are central regulatory lipids that direct membrane function in eukaryotic cells. Understanding how their synthesis is regulated is crucial to revealing these lipids' role in health and disease. PPIn are derived from the major structural lipid, phosphatidylinositol (PI). However, although the distribution of most PPIn has been characterized, the subcellular localization of PI available for PPIn synthesis is not known. Here, we used several orthogonal approaches to map the subcellular distribution of PI, including localizing exogenous fluorescent PI, as well as detecting lipid conversion products of endogenous PI after acute chemogenetic activation of PI-specific phospholipase and 4-kinase. We report that PI is broadly distributed throughout intracellular membrane compartments. However, there is a surprising lack of PI in the plasma membrane compared with the PPIn. These experiments implicate regulation of PI supply to the plasma membrane, as opposed to regulation of PPIn-kinases, as crucial to the control of PPIn synthesis and function at the PM.**

## Introduction

The polyphosphoinositides (PPIn) are crucial regulatory lipids in eukaryotic physiology. They direct protein localization and/or activation on the cytosolic face of membranes, thereby controlling myriad cellular processes such as membrane traffic, lipid exchange, ion transport, cell signaling, and cytoskeletal dynamics (Balla, 2013; Dickson and Hille, 2019). Structurally, PPIn consist of phosphorylated derivatives of the major glycerophospholipid, phosphatidylinositol (PI). The enzymology of PPIn synthesis and turnover is well understood, and an important endeavor in cell biology seeks to understand the control of cellular physiology in both health and disease by understanding regulation of PPIn enzymes.

When their relative abundance is considered, the PPIn have an outsized role in membrane function; they account for only ~2–10% of inositol lipid, with the remainder being PI (Anderson et al., 2013; Traynor-Kaplan et al., 2017). PI itself accounts for ~10% of total phospholipid (Vance, 2015), so it follows that PPIn constitute <1% of total phospholipid. On the other hand, PI is an important structural component of membranes, and its synthesis probably accounts for positive effects of dietary inositol supplementation, as opposed to effects on quantitatively minor PPIn (Michell, 2018).

PI is synthesized on the cytosolic face of the ER (Bochud and Conzelmann, 2015), from where some is "flopped" to the luminal leaflet for the synthesis of glycosylphosphatidylinositol-linked proteins (Vishwakarma et al., 2005). The remaining PI has been shown to be distributed fairly evenly across most organelle membranes by subcellular fractionation (Vance, 2015). However, in such studies, the fraction of PI in the cytosolic face of the membrane (and thus available for PPIn synthesis) is not known. Furthermore, the process of subcellular fractionation exposes membranes to phosphatases that can remove labile PPIn phosphate monoesters, consequently overestimating PI and underestimating PPIn. Therefore, it is currently unclear how much PI is available for PPIn synthesis in cytosolic membrane leaflets—and hence whether the crucial regulatory step is the control of PI kinase activity or the supply of PI substrate.

A prominent example of this problem is at the plasma membrane (PM), which contains the majority of PI 4,5-bisphosphate ($PI(4,5)P_2$) and the largest share of PI4P (Hammond and Balla, 2015). Although these lipids are ≤1% of total cellular phospholipid, they are specifically enriched at the PM, whereas the much more abundant PI is not, so PPIn likely account for a larger share of the PM inositol lipid. How big a share? Estimates of the fraction of cellular phospholipids at the PM vary by two orders of magnitude, from 0.5% (Schmick et al., 2014) to 50% (Lange et al., 1989); assuming an intermediate fraction of 14% (Griffiths et al., 1989), $PI(4,5)P_2$ and PI4P may therefore be

Department of Cell Biology, University of Pittsburgh School of Medicine, Pittsburgh, PA.

Correspondence to Gerry R.V. Hammond: ghammond@pitt.edu; B.D. Goulden's present address is Department of Cell Biology, Johns Hopkins University School of Medicine, Baltimore, MD.

~7% of PM phospholipid, not far from the roughly 10% PI assumed if it was uniformly distributed among organelles including the PM. This fits with measurements of red blood cells, which only contain PM, where PI, PI4P, and PI(4,5)$P_2$ are present in approximately equal quantities (Ferrell and Huestis, 1984). It also fits with observations from isolated cardiomyocyte PM, which has a limited supply of PI for PPIn resynthesis after stimulation of PLC (Nasuhoglu et al., 2002). In general though, there is scant evidence as to the relative abundance of PI in various organelles that is actually available for PPIn synthesis.

In this study, we applied a series of distinct approaches to map the subcellular distribution of PI in intact, living cells. These included loading cells with fluorescent PI derivatives and probing cells for DAG and PI4P after inducing acute conversion of endogenous PI to these lipids. While each approach carries significant caveats, collectively our results illustrate a wide subcellular distribution of PI, but a surprising lack at the PM.

## Results

### Intracellular distribution of exogenous fluorescent PI

Lipids with fluorescent fatty acids have long been used as tracers for the traffic, metabolism, and steady-state distribution of native lipids (Lipsky and Pagano, 1985; Struck and Pagano, 1980). In these experiments, exogenous lipids are applied to cells as either liposomes or via serum albumin carriers, from where they spontaneously incorporate into the exoplasmic leaflet of the PM. From here, the lipids follow the same cellular fate as the native molecules: they are either endocytosed or else are flipped by native translocases. The flipped lipids may then traffic from the cytosolic leaflet of the PM to other organelles via vesicular and nonvesicular pathways, being metabolized along the way (Pagano et al., 1983). We attempted this approach with inositol lipids, using commercially available TopFluor fatty acid conjugated lipids. We reasoned this was a potentially viable approach, since native PI translocase and/or scramblase activities have been reported in mammalian cells, albeit with weaker activity compared with that of aminophospholipid translocases (Bütikofer et al., 1990; Wang et al., 2018).

We loaded COS-7 green monkey fibroblasts with TopFluor lipids using BSA for 15 min before back-extracting the outer PM leaflet with excess BSA (Fig. 1 A). As a positive control, we loaded cells with TopFluor-phosphatidylserine (PS). This lipid rapidly incorporated into the PM and intracellular vesicles (Fig. 1 A, top panel). It could not be back-extracted since the PS is rapidly flipped to the inner leaflet by endogenous aminophospholipid translocases (Fig. 1 A), exactly as described previously (Kay et al., 2012). On the other hand, TopFluor–PI(4,5)$P_2$ intensely labeled the PM but was almost entirely back-extracted, leaving only a punctate signal that presumably corresponded to endocytosed lipid in the exoplasmic leaflet (Fig. 1 A, middle panel). The majority of TopFluor-PI was also back-extracted from the PM. However, in addition to the internalized vesicles, a substantial intracellular labeling appeared during loading that remained after back-extraction (Fig. 1 A, bottom panel). This suggested that PI was indeed flipped into the inner PM leaflet but, in

contrast to PS, was subsequently transported rapidly to other organelles.

Previous work has shown a similar, though much more rapid, translocation of NBD-labeled PI from the cell surface to intracellular membranes in 3T3 preadipocytes, though not in Chinese hamster ovary cells (Ting and Pagano, 1990). That rapid translocation was due to exoplasmic PLC activity in 3T3 cells that cleaved PI into the rapidly flip-flopping lipid DAG. Therefore, we extracted lipids from our COS-7 cells loaded with TopFluor-PI after back-extraction for analysis by TLC; we observed no such metabolism of PI, which migrated identically to the TopFluor-PI standard and was well resolved from TopFluor-DAG (Fig. 1 B). Thus, COS-7 cells appeared to traffic exogenous fluorescent PI molecules from the PM and distribute them to intracellular membranes intact.

To identify these intracellular membranes, we performed high-resolution confocal microscopy of live cells expressing fluorescent protein-conjugated markers of the most abundant intracellular membranes. With this approach, we could detect clear enrichment of the lipid at Golgi membranes (Fig. 1 C and Fig. S1), the ER (Fig. 1 D and Fig. S1), and mitochondria (Fig. 1 E and Fig. S1). Surprisingly, relatively little TopFluor-PI was observed at the PM, in stark contrast to the observations with TopFluor-PS (Fig. 1 A). To the extent that TopFluor-PI traffic and steady-state distribution mirrors natural PI, this implies that PI is widely distributed in the cell, though notably absent from the PM, one of the most active compartments of phosphoinositide metabolism and function. However, these experiments still bear the significant caveat that the exogenous, derivatized PI may not reflect the endogenous distribution and traffic of native PI. They also do not furnish information on the transbilayer distribution of these lipids within the labeled organelles. For these reasons, we next turned our attention to approaches that could probe for the presence of endogenous PI in the cytoplasmic face of organelle membranes.

### An acutely activatable PI-PLC to probe organelle PI content

In lieu of a bona fide PI biosensor for use in living cells, the PI-specific PLC from *Listeria monocytogenes* can be used as an indirect probe: the PI-PLC converts PI to inositol phosphate and DAG, the latter of which can then be detected with the selective, high-affinity C1ab domain from PKD1 (Kim et al., 2011). The PI-PLC retains high activity despite its largely cytosolic localization, revealing highly dynamic DAG-containing structures that are not easily attributable to specific organelles of origin. However, prolonged exposure of cells to a degradative enzyme like a phospholipase risks extensive damage to membranes and activation of containment and repair processes. In other words, the DAG distribution may not closely reflect the intracellular origin of the PI, which may have been generated several hours before imaging in these transfection experiments (Kim et al., 2011).

To circumvent this problem, we attempted to make an acutely activatable PI-PLC. Recruitment of lipid-modifying enzymes to membranes is one commonly used approach to accomplish this (DeRose et al., 2013), but the high basal activity of cytoplasmic PI-PLC (Kim et al., 2011) precluded this approach. We attempted to make mutants of *Bacillus cereus* PI-PLC with

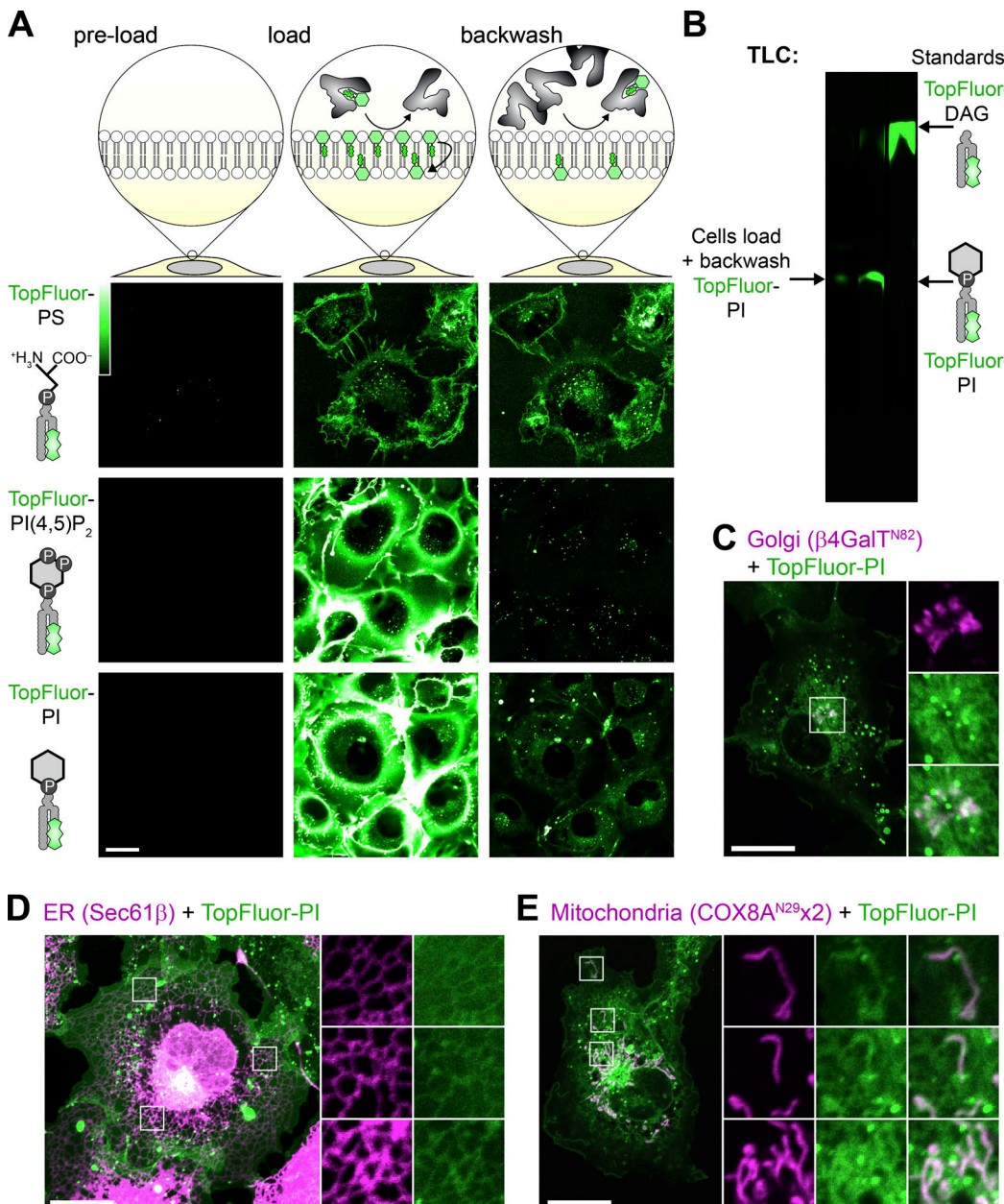

Figure 1. **Fluorescent PI is not enriched at the PM. (A)** Internalization of fluorescent lipids. COS-7 cells were loaded at 37°C with the indicated acyl-conjugated TopFluor lipids complexed with BSA. After 15 min, a 20-fold excess of un-complexed BSA was used to back-extract lipid remaining in the outer PM leaflet. **(B)** Loaded TopFluor-PI is not metabolized. COS-7 cells were loaded or loaded and back-extracted as in A. Lipids were then extracted and resolved by TLC. **(C–E)** Loaded TopFluor-PI labels the Golgi (C), ER (D), and mitochondria (E). COS-7 cells that had been transfected with mCherry-β4-GalT$^{N82}$ (C), iRFP-Sec61β (D), or COX8A$^{N29}$x2-mCherry (E) were loaded and back-extracted as in A. Scale bar = 20 μm in all panels; insets are 10.9 μm in C or 7.3 μm in D and E. Data are representative of three or more experiments.

reduced (but not abolished) catalytic activity based on previously reported activity of the recombinant enzyme (Gässler et al., 1997). However, none of the mutants we screened retained sufficient activity to yield DAG production upon recruitment while also showing low basal activity before recruitment. We also attempted to incorporate caged lysine into the *L. monocytogenes* enzyme using unnatural amino acids (Courtney and Deiters, 2018), though we were unable to obtain expression of the mutated enzyme.

Finally, on inspecting the crystal structure of the *L. monocytogenes* enzyme, we noted that the protein consisted of two well-defined amino and carboxy-terminal lobes with a single linking loop and also possessed amino and carboxy termini that were close to one another (Moser et al., 1997). We therefore reasoned that the enzyme could be expressed as separate lobes that could be induced to form a functional enzyme after chemically induced dimerization of FKBP (FK506-binding protein) and FRB (FKBP12 and rapamycin binding domain of mTor)

fused to the amino and carboxy-terminal domains, respectively (Fig. 2 A).

Expression of TagBFP2-FKBP-PLC$^{N187}$ and PLC$^{C100}$-FRB-iRFP showed both enzymes distributed in the cytoplasm, as observed with the full-length enzyme (Kim et al., 2011); no change in localization was observed when dimerization was induced with rapamycin (see micrographs in Fig. 2 A). However, a GFP-C1ab DAG biosensor expressed in the same cells began to show an intense juxtanuclear accumulation and many small punctate structures in the cytoplasm (Fig. 2 B). We identified many of these structures through colocalization with coexpressed organelle markers (Fig. 2 B). The juxtanuclear accumulation occurred in and around the Golgi membranes marked by ManII$^{N102}$. Many of the cytoplasmic puncta showed colocalization with Rab5 or LAMP1 as markers of the early and endo/lysosomal compartments, respectively. Surprisingly, and in contrast to results with TopFluor-PI, we did not see substantial accumulation in the ER or mitochondria, nor peroxisomes. A small, statistically significant change in localization of C1ab fluorescence with peroxisomes and the ER was observed, though these may simply be due to chance overlap given the large increases in the number of cytoplasmic C1ab-labeled puncta. None seemed to precisely correspond to these organelles as they did for the Golgi and endosomes.

The most parsimonious explanation for our failure to observe PLC-mediated DAG generation at the ER and mitochondria would be that little PI remains in the cytosolic leaflets of these organelle membranes. However, it is also possible that the cytosolic enzyme is not active on these membranes, or that the C1ab probe does not efficiently recognize DAG in these membrane contexts. Moreover, substantial recruitment of C1ab to DAG-replete membranes like the Golgi may also deplete the probe available to detect smaller pools elsewhere. To more directly probe for the presence of PI substrate, we targeted the amino-terminal lobe of PI-PLC to the outer mitochondrial membrane by fusing it to the amino-terminal 31 amino acids of AKAP1 (Ma and Taylor, 2002), with the aim of recruiting the carboxy-terminal half to reconstitute the enzyme at the organelle surface (Fig. 2 C). Indeed, we could see robust recruitment of PLC$^{C100}$-FRB-iRFP with a time constant of ~3 min, which now yielded a substantial increase in C1ab labeling of the mitochondria with similar kinetics (Fig. S2, B and D). Interestingly, although DAG appeared initially throughout the mitochondrial surface, the labeling rapidly resolved into small puncta. We do not know the nature of this relocalization of DAG, which could conceivably be phase separation or traffic of the accumulated lipid. It may, however, explain the failure to observe substantial labeling of the mitochondria by the cytosolic enzyme in our experiments (Fig. 2 B) or previous studies (Kim et al., 2011).

We attempted similar experiments with the ER. However, fusion of PI-PLC$^{N187}$ to the amino-terminal domain of STIM1 exhibited a mis-localized, punctate distribution in addition to the normal ER morphology, whereas fusion of PLC$^{C100}$ to the SAC1 carboxy-terminal transmembrane domains failed to recruit cytosolic PI-PLC$^{N187}$ (unpublished data). Therefore, we were unable to definitively probe the cytosolic face of the ER with this approach.

We also produced a PM-targeted PI-PLC$^{N187}$ by fusion to the myristoylated and palmitoylated amino-terminal domain of Lyn kinase (Fig. 3 A). Dimerization with rapamycin induced translocation of PI-PLC$^{C100}$ to the PM that was largely complete in 1 min (Fig. S2, B and C) and the formation of a few C1ab-labeled puncta in these cells, visible by total internal reflection fluorescence microscopy (TIRFM). Nonetheless, the overall increase in DAG was small and transient. In contrast, activation of PI(4,5) P$_2$–specific PLCβ by activation of either endogenous purinergic or overexpressed muscarinic M3 receptors caused a much larger increase in DAG (Fig. 3 A). This result would be consistent with the notion that in the PM, levels of PI are much lower than those of PI(4,5)P$_2$; hydrolysis of a relatively small fraction of PI(4,5)P$_2$ induced by ATP still produces a much greater increase in DAG than PI-PLC–mediated hydrolysis of PM PI.

An alternative explanation to these data could be that the Lyn amino-terminal fusion does not orient the split PI-PLC in an orientation conducive to activity in the PM; the few DAG puncta that we observed may instead be produced on organelle membranes that happen to approach close enough to the PM for the recombined PI-PLC to hydrolyze their PI. Therefore, in order to test whether PM-targeted PI-PLC indeed generated DAG at the PM, we sought an alternative route to detect PM-localized DAG. To this end, we took advantage of the fact that translocation of the ER-localized PI/phosphatidic acid transfer protein Nir2 requires PM DAG to translocate to ER-PM contact sites (Kim et al., 2015). As shown in Fig. 3 B, dimerization of split PI-PLC at the PM induced translocation of Nir2 to puncta as seen in TIRFM; this occurred to virtually the same extent as with activation of endogenous PLCβ with ATP. Although there were small differences in the kinetics of the translocation, no significant difference between the two stimuli was detected (Fig. 3 B). Together with our observations with the C1ab DAG sensor, these data imply that sufficient PI is present at the PM that, when converted to DAG, can recruit Nir2; but it is far less than the quantity produced when hydrolysis of PI(4,5)P$_2$ is activated by endogenous PLCβ.

We also tested whether acute activation of PI-PLC reduced PI levels sufficiently to reduce levels of phosphoinositides, as was previously reported for constitutive overexpression of the intact enzyme (Kim et al., 2011). Notably, the large increase in C1ab after PI-PLC activation correlated with a partial depletion of the Golgi-associated PI4P biosensor P4M (Fig. 4 A). However, within 15 min, we could not detect decreases in endosomal association of PI3P biosensor FYVE-EEA1 (Fig. 4 A) or PM PI(4,5)P$_2$ biosensor Tubby$_c$ (Fig. 4 B). This could be interpreted as a simple failure to sufficiently deplete the endosome or PM-associated PI levels. Alternatively, it would be consistent with the PI pools used in synthesis of these lipids being derived from other, non–PLC–accessible sources that are not necessarily associated with these organelles. Unexpectedly, we observed a small (~10%) but significant increase in the PM PI4P biosensor P4M (Fig. 4 B). It is possible that this results from a paradoxical increased supply of PI to the PM, caused by the PI-PLC–induced translocation of Nir2 that we reported in Fig. 3 B. This would require the endogenous PI 4-OH kinase (PI4K) to consume Nir2-delivered PI before the PM-targeted split PI-PLC, which would also be consistent

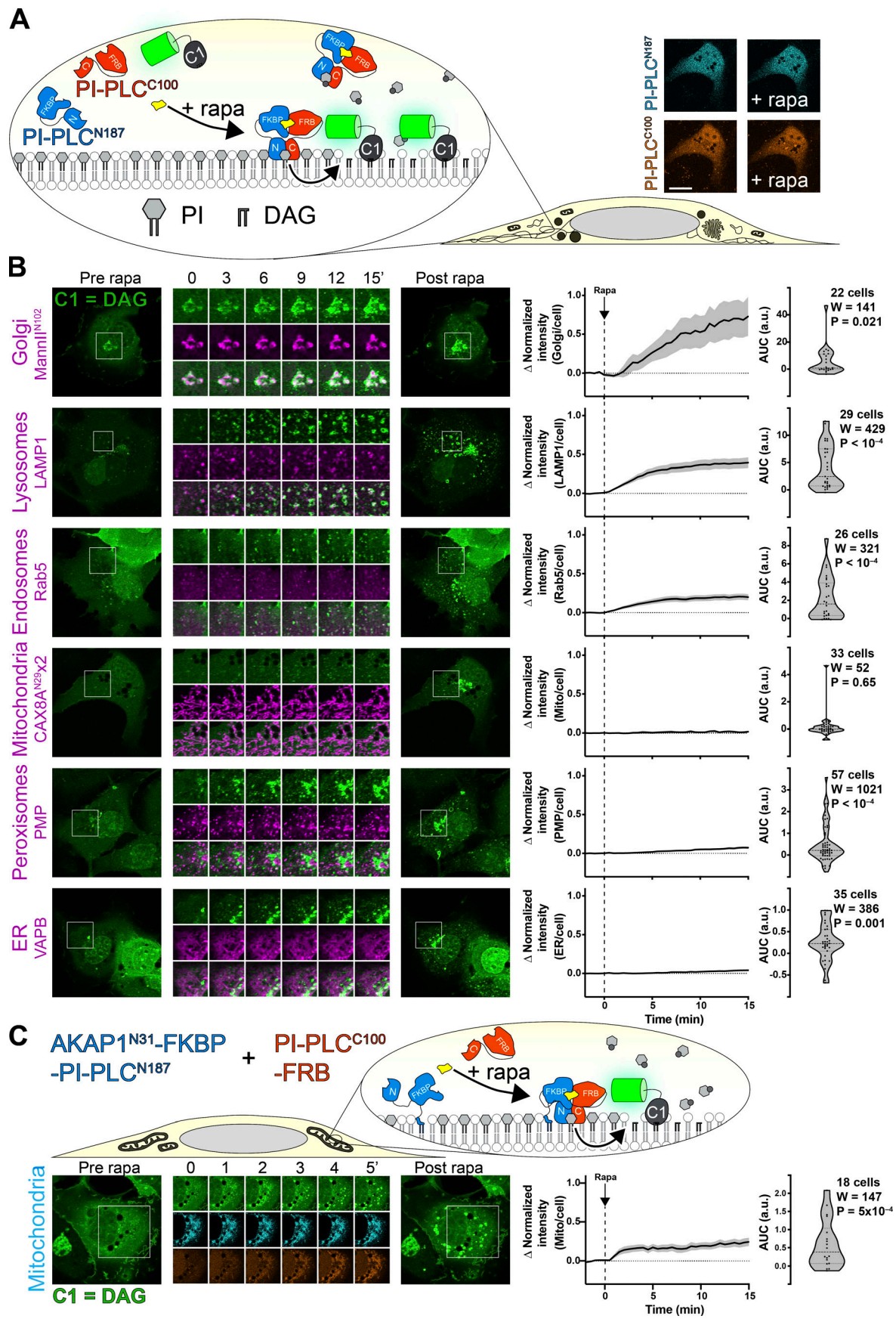

Figure 2. **Chemically induced dimerization of a split PI-PLC induces intracellular accumulation of DAG. (A)** Experimental setup: rapamycin-induced dimerization of FKBP-fused amino-terminal 187 residues of PI-PLC (cyan) with the FRB-fused carboxy-terminal 100 residues (orange) reconstitutes the active enzyme, though no visible change in the cytosolic localization of the TagBFP2/iRFP-fused enzyme fragments is observed (confocal images at right; scale bar = 20 µm). **(B)** DAG accumulation on cytosolic leaflets of intracellular organelles, with the greatest increases in the Golgi and endosomal/lysosomal compartments. Zero or comparatively minor changes were observed in mitochondria, peroxisomes, or the ER. Cells were expressing GFP-PKD1-C1ab to detect DAG, the indicated organelle markers fused to mCherry or mKO (magenta), FKBP-PI-PLC$^{N187}$, and FRB-PI-PLC$^{C100}$ (not shown); they were treated with 1 µM rapamycin at time 0. Inset regions are 15 µm and serve as scale bars. **(C)** Anchoring PI-PLC$^{N187}$ to the mitochondrial outer membrane shows transient accumulation of DAG after recruitment of PI-PLC$^{C100}$. Cells were transfected as in B, but with AKAP$^{N31}$—fused PI-PLC$^{N187}$ replacing the unanchored version in B. Inset = 30 µm and serves as scale bar. In both B and C, the curves at right show the mean change in C1ab reporter intensity at each compartment, with SEM shaded. The violin plots show AUC, with the number of cells (pooled across three to six independent experiments), the sum of signed ranks (W), and P value from a two-tailed Wilcoxon signed rank test compared to a null hypothesis AUC value of 0. mito, mitochondria.

with the failure of PI-PLC to deplete PI4P or PI(4,5)P$_2$ in these experiments.

Overall, these data demonstrate that PI is widely distributed in intracellular cytosolic membranes, though notably absent from the PM. These data are largely consistent with our observations of fluorescent PI redistribution, though there were a number of differences, notably our failure to detect ER-associated PI. Therefore, we sought an alternative, corroborative approach.

**Acute conversion to PI4P as a probe for PI**

We reasoned that we could also detect the presence of PI via conversion to PI4P after recruitment of PI4K, in a similar approach to detecting DAG derived from acutely reconstituted PI-PLC. To this end, we selected PI4KIIIα (encoded by *PI4KA*, referred to hereafter as PI4KA). The enzyme exists as a large, 700-kD multi-subunit complex (Lees et al., 2017b; Dornan et al., 2018); however, the isolated carboxy-terminal fragment containing the helical and catalytic domain (PI4KA$^{C1001}$) retains activity and can complement hepatitis C proliferation in PI4KA knockdown cells (Harak et al., 2014). Therefore, we fused this fragment to FKBP to facilitate chemically induced dimerization with FRB, targeted to distinct organelle membranes (Fig. 5 A). A catalytically inactive Asp$^{1957}$ to Ala mutant served as a negative control. PI4P was detected with highly selective and unbiased probes for PI4P, namely the relatively low affinity P4M domain (Hammond et al., 2014) and the higher affinity P4C domain (Weber et al., 2014; Luo et al., 2015) from *Legionella pneumophila* effectors SidM and SidC, respectively.

Kinetically, recruitment of FKBP-PI4KA$^{C1001}$ occurred with a time constant of 1–2 min to each organelle, with any PI4P biosensor recruitment occurring over a similar or slightly slower time constant (Fig. S3). In terms of magnitude, recruitment of PI4KA$^{C1001}$ to the Golgi induced a rapid and dramatic increase in PI4P levels at the Golgi (Fig. 5 B), consistent with our observations with DAG after PI-PLC activation (Fig. 2 B). Recruitment to Rab5- or LAMP1-positive endosomes and lysosomes produced a small but significant increase in PI4P, though this was less marked than the increases in DAG after PI-PLC activation (Fig. 2 B). On the other hand, we could observe a dramatic increase in PI4P on the mitochondrial outer membrane when recruiting PI4KA$^{C1001}$ and a marked increase at peroxisomal membranes (Fig. 5 B). Strangely, we noticed that in addition to producing PI4P in structures colocalizing with FRB-peroxisomal membrane protein 2 (PMP), an increase was also observed at a

compartment with morphology consistent with mitochondria. We do not know if this is due to transfer of PI4P produced at peroxisomes to the mitochondrial outer membrane, perhaps via contact sites (Valm et al., 2017), or else direct synthesis of PI4P at mitochondrial membranes by a small pool of mis-targeted PMP.

We could barely detect PI4P synthesis at the ER after PI4-KA$^{C1001}$ recruitment (Fig. 5 B). We reasoned that this was likely due to the presence of SAC1, a highly active PI4P phosphatase present throughout the ER (Zewe et al., 2018). Inhibition of SAC1 with peroxide causes rapid accumulation of PI4P at the ER, which has been interpreted as being due to PI4P transfer from other organelles (Zewe et al., 2018). We therefore reasoned that recruitment of PI4KA$^{C1001}$ during inhibition of SAC1 would cause a further increase in PI4P levels due to conversion of any PI already present in the ER. Indeed, this is exactly what we observed: there was a substantial and highly significant increase in ER PI4P accumulation when active PI4KA$^{C1001}$ was recruited relative to the inactive control (Fig. 5 C). Therefore, we could corroborate the presence of PI in the ER suggested by the accumulation of TopFluor-PI there (Fig. 1 D).

In contrast, we could detect no increases in PI4P at the PM by TIRFM after recruiting PI4KA$^{C1001}$ (Fig. 6 A), despite efficient recruitment within 1 min (Fig. S3, B and C), consistent with our observations with PI-PLC (Fig. 3) and the distribution of TopFluor-PI (Fig. 1). We also saw no increase in PI(4,5)P$_2$ detected with the low affinity Tubby c-terminal domain mutant, Tubby$_c^{R332H}$ (Quinn et al., 2008), ruling out conversion of extra PI4P to this lipid (Fig. 6 A). An obvious interpretation of this result is that there is little PI resident in the PM available for conversion to PI4P (or to PI(4,5)P$_2$). However, we wanted to rule out other interpretations—specifically, that the PI4KA catalytic domain may be "biologically insufficient" to phosphorylate PI in the PM without assistance from PI transfer proteins (Grabon et al., 2015). To this end, we devised an experiment whereby the endogenous PM-associated PI4KA activity could be replaced with an alternative activity. For this purpose, we selected PI4KIIIβ (PI4KB), the isoform usually associated with PI4P synthesis at the Golgi (Balla and Balla, 2006). We made an FKBP fusion of this enzyme; recruitment to mitochondria occurred within 1 min and caused accumulation of PI4P in this membrane with a time course of ~11 min (Fig. 6 B and Fig. S4), demonstrating the fusion was active. On the other hand, as we observed for FKBP-PI4KA$^{C1001}$, no increases in PI4P were observed after recruitment of FKBP-PI4KB to the PM (which occurred within 1 min; Fig. S4) relative to an FKBP-only control (Fig. 6 B).

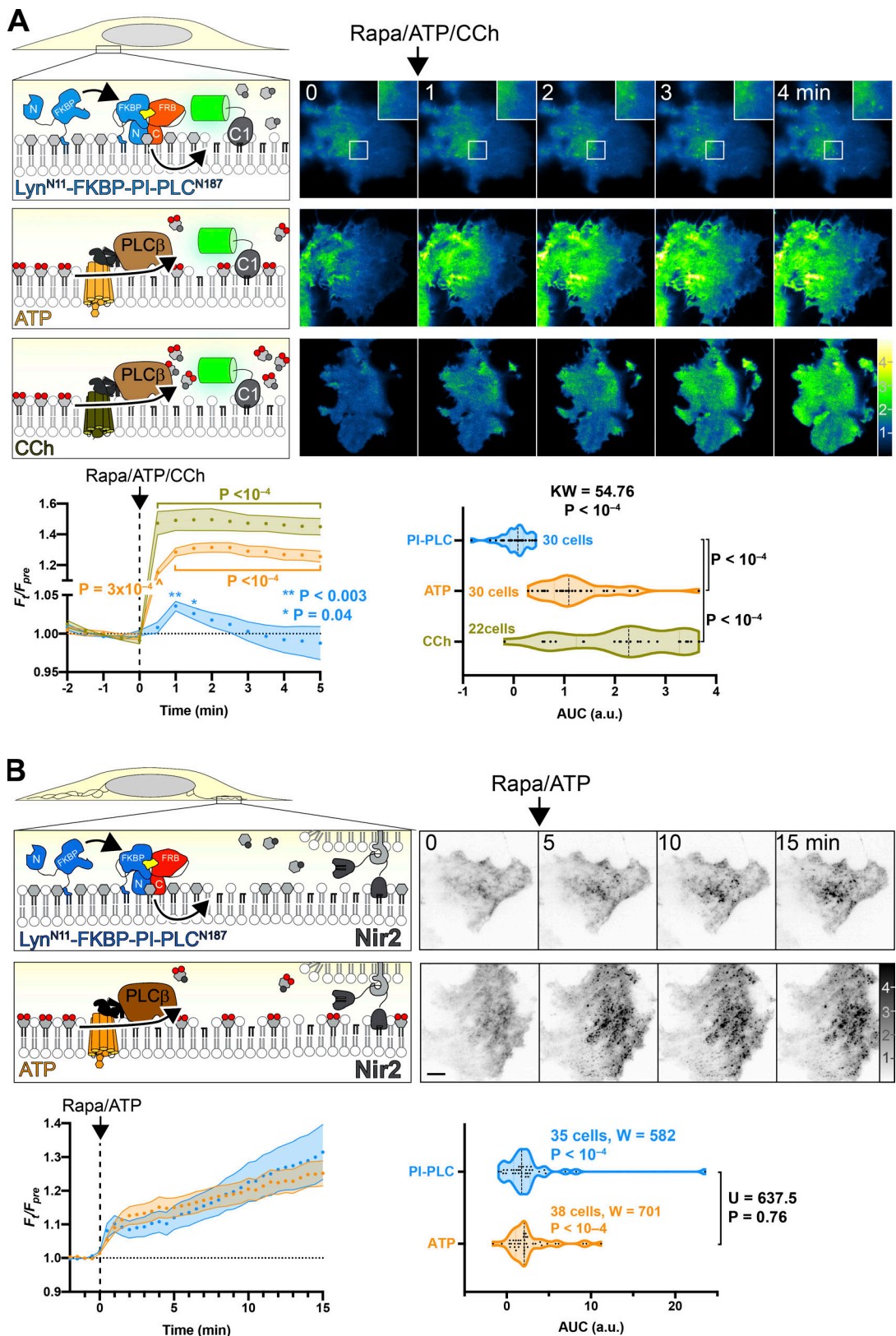

Figure 3.   **Very little PI can be converted to DAG at the PM. (A)** PM-specific dimerization of split PI-PLC induces very little DAG compared with PI(4,5)P$_2$-specific PLC. Cells were transfected with PI-PLC$^{C100}$-FRB, PM-targeted Lyn$^{N11}$-FKBP-PI-PLC$^{N100}$, and GFP-PKD1-C1ab to detect DAG. Rapamycin was used to induce PI-PLC reconstitution at the PM, or else endogenous PLCβ was activated by G$_q$-coupled agonist ATP (to activate endogenous P2Y receptors) or carbachol (CCh; to activate overexpressed muscarinic M3 receptors). TIRFM images are color-coded to represent fluorescence intensity relative to prestimulus levels ($F_t/F_{pre}$) as indicated. Inset region is 5 µm and serves as scale bar for images in A. The line graphs show mean $F_t/F_{pre}$ with SEM shaded; P values are

derived from Dunn's multiple comparison test compared to time 0, following Friedman's test (Friedman statistic = 49.16, PI-PLC, P = 0.045; 497.0, ATP, P < 10$^{-4}$; 282.3, CCh; P < 10$^{-4}$). Where not indicated, the P value from Dunn's test is >0.05. The violin plots show AUC analysis of the line graphs with the number of cells from three independent experiments, with results of a Kruskal-Wallis test and P values from a post hoc Dunn's multiple comparison test indicated. **(B)** Split PI-PLC induces translocation of Nir2 to ER-PM contact sites. Cells were transfected with PM-targeted split PI-PLC and stimulated with rapamycin or ATP as in A. Translocation of GFP-Nir2 was recorded; images show representative TIRFM images with fluorescence normalized to prestimulus levels ($F_t/F_{pre}$). Scale bar = 10 μm. The line graphs show mean $F_t/F_{pre}$ with SEM shaded; the violin plots show AUC analysis of the line graphs with the number of cells from three independent experiments indicated, along with results of Wilcoxon signed rank test comparing each population to a hypothesized AUC of 0, as well as a Mann-Whitney $U$ test comparing the differences between AUC after ATP or rapamycin stimulation.

We next sought to demonstrate that PI4KB can intrinsically be active at the PM, ruling out biological insufficiency of PI4KB or that the enzyme has a preference for acyl chains found in the Golgi but not PM PI. One situation in which a rapid increase in PM PI4P and PI(4,5)P$_2$ synthesis occurs is after recovery from PLCβ-mediated depletion of these lipids following muscarinic M3 acetylcholine receptor activation (Willars et al., 1998). We overexpressed M3 receptors in COS-7 cells expressing the Tubby c-terminal domain as a PI(4,5)P$_2$ reporter (Fig. 6 C). Cells were stimulated with carbachol for 2 min to activate PLCβ, and then the muscarinic antagonist atropine was added to shut off the PLC and facilitate PI(4,5)P$_2$ resynthesis via PI4P (Nakanishi et al., 1995). Control cells rapidly resynthesized PM PI(4,5)P$_2$ within

4 min (Fig. 6 C). We then repeated this experiment while blocking endogenous PI4KA activity with the highly potent and selective inhibitor A1 (Bojjireddy et al., 2014); very little resynthesis of PI(4,5)P$_2$ occurs under these conditions (Fig. 6 C). Finally, we repeated the A1 treatment in cells in which A1-resistant FKBP-PI4KB was recruited to the PM: This led to a substantial recovery of PM PI(4,5)P$_2$ synthesis, though this was slower than in controls and incomplete in the 15-min period of the experimental recovery (Fig. 6 C). Nonetheless, clear activity of PI4KB could be demonstrated in the PM, strongly arguing that a failure to further increase PI4P (or PI(4,5)P$_2$) levels in unstimulated cells is due to a scarcity of PI available for conversion.

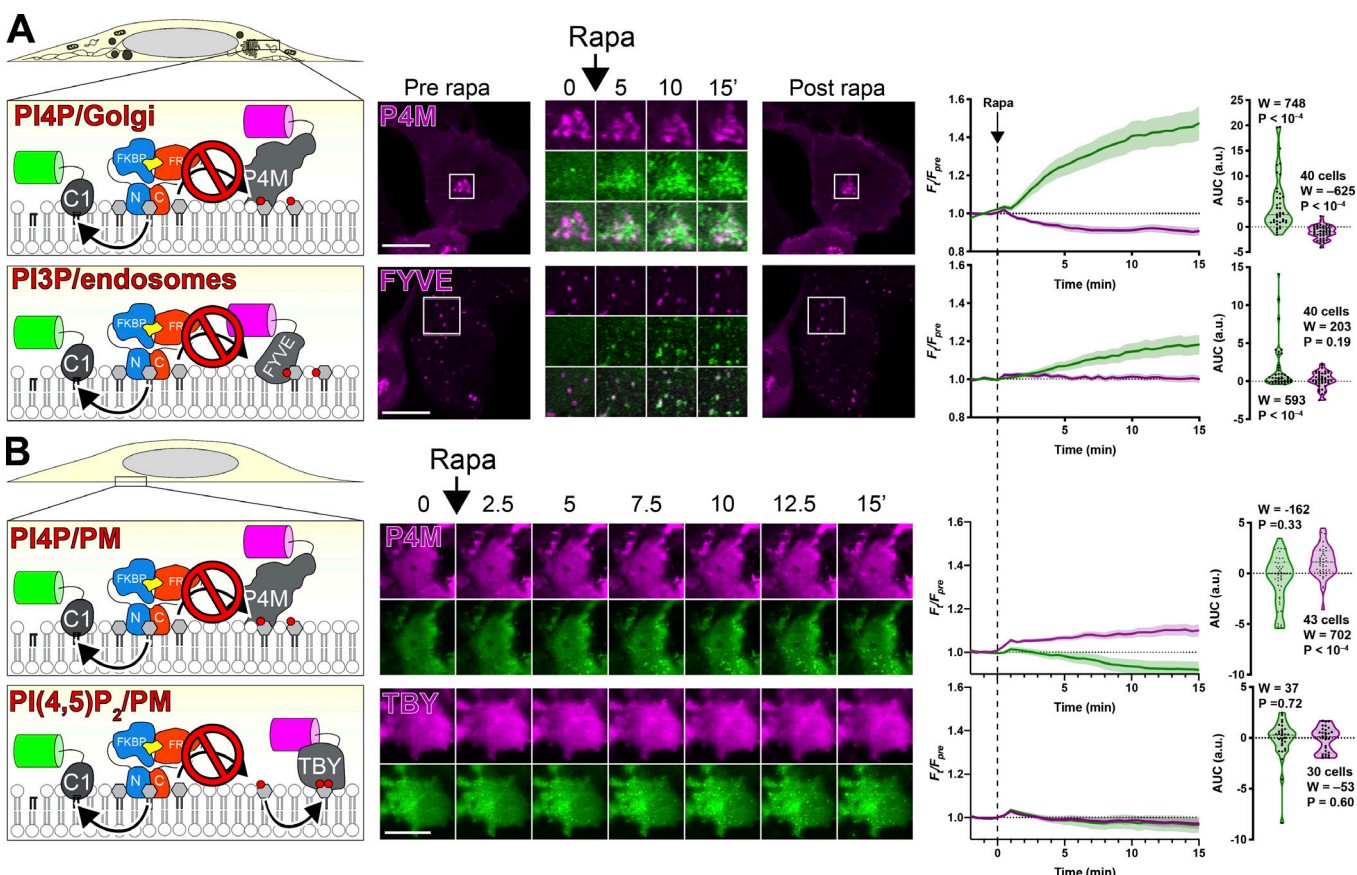

Figure 4. **PI-PLC leads to little or no depletion of PPIn on Golgi and endosomes or PM. (A and B)** Cells were transfected with PI-PLC$^{C100}$-FRB, FKBP-PI-PLC$^{N100}$ (or PM-targeted Lyn$^{N11}$-FKBP-PI-PLC$^{N100}$), and GFP-PKD1-C1ab to detect DAG, along with the indicated PPIn biosensor. Dimerization and activation of PI-PLC was induced with 1 μM rapamycin as indicated. Images are confocal sections (A) or TIRFM (B). Scale bars = 20 μm. Inset = 10 μm for P4M and 15 μm for FYVE. Line graphs show the change in compartment-specific fluorescence of C1ab (green) or the PPIn biosensor (magenta) normalized to prerapamycin levels ($F_{pre}$). The violin plots show AUC, with the number of cells (pooled across three independent experiments), the sum of signed ranks (W), and P value from a two-tailed Wilcoxon signed rank test compared to a null hypothesis AUC value of 0 (with a baseline of 1).

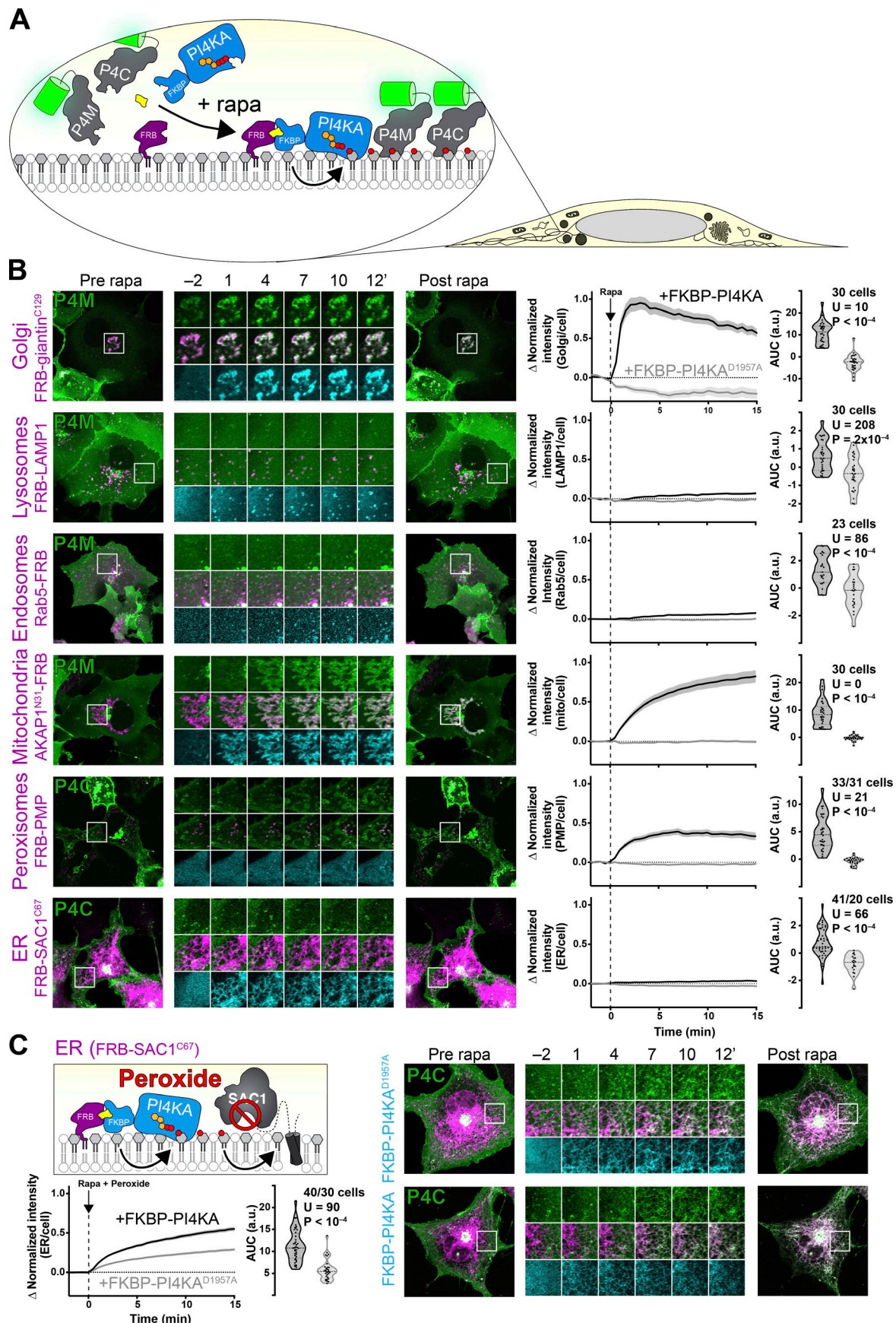

Figure 5.  **Compartment-specific recruitment of PI4KA reveals intracellular PI pools. (A)** Experimental setup: rapamycin-induced recruitment of FKBP-PI4KA[N1001] by dimerization with compartment-specific FRB to induce PI4P synthesis from endogenous PI, revealed with GFP-P4M or -P4C. **(B)** PI4P accumulation on cytosolic leaflets of intracellular organelles, with the greatest increases in the Golgi and mitochondria, with barely detectable increases in endosomes and ER. Peroxisomes show a large increase, but PI4P intensity also occurs outside of PMP-marked compartments. Cells expressing GFP-P4M or -P4C (green), compartment-specific FRB (magenta), and FKBP-PI4KA[N1001] (cyan) as indicated were treated with 1 μM rapamycin at time 0. Inset regions are 15 μm and serve as scale bars. The line graphs at right show the mean change in PI4P reporter intensity at each compartment, with SEM shaded. The violin plots show AUC analysis of the curves, with the number of cells (pooled across two to four independent experiments), Mann-Whitney $U$ statistic, and P value from a two-tailed test. **(C)** Peroxide-mediated inhibition of ER-associated SAC1 PI4P phosphatase reveals an endogenous pool of PI that can be converted to PI4P. Experiment and data are identical to B, except cells were treated with 500 μM hydrogen peroxide in addition to rapamycin. mito, mitochondria.

## Discussion

In this paper, we performed experiments to map the subcellular distribution of PI in intact, living cells. We were particularly interested with respect to organelles that contain PPIn, since the availability of PI substrate for PPIn synthesis has important implications for the regulation of PPIn abundance and downstream physiology. We took three complimentary approaches: localization of an exogenous, fluorescent PI (Fig. 1); localization of DAG, converted from PI by an acutely activated PI-PLC (Fig. 2, Fig. 3, and Fig. 4); and localization of PI4P after acute conversion of PI by a membrane-recruited PI4K (Fig. 5 and Fig. 6). Broadly speaking, these approaches demonstrated the presence of PI in multiple intracellular membranes, but a relatively small amount (compared with PPIn) at the PM (see the summary of results in Table 1).

Each of the three approaches has significant caveats. Optical imaging of fluorescent lipids fails to give information as to the bilayer distribution, so it does not necessarily reflect the PI available for conversion to PPIn by cytosolic PI kinases. Furthermore, the fluorescent moiety may disrupt the traffic and distribution of the exogenous lipid; for example, whereas TopFluor-labeled PS reflects closely the distribution of endogenous PS (Kay et al., 2012), NBD-labeled PS fails to show enrichment at the cytosolic face of the PM (Martin and Pagano, 1987).

On the other hand, acute conversion of PI to DAG or PI4P by acute chemogenetic activation of enzymes strictly reports endogenous, cytosolic leaflet-resident PI. Nevertheless, there are also substantial confounds to these experiments. Breakdown of a large fraction of phospholipid in a membrane to the much less polar DAG may dramatically alter bilayer properties (Alwarawrah et al., 2016). Moreover, DAG can readily flop to the exoplasmic leaflet (Bai and Pagano, 1997), or else be metabolized by DAG kinase, acyltransferases, or lipases. Each process could significantly limit translocation of the C1ab domain probe, perhaps explaining the transient DAG accumulation observed at mitochondria before the signal resolves into a more punctate distribution (Fig. 2 C). Likewise, acutely induced PI4P could be subject to phosphorylation or dephosphorylation by PPIn enzymes or else sequestration or transport by, for example, the OSBP-related family of proteins (Olkkonen, 2015). Indeed, the presence of the PI4P phosphatase Sac2 and the PI4P transfer protein OSBP at endosomes (Hsu et al., 2015; Nakatsu et al., 2015; Dong et al., 2016) may explain our failure to induce significant PI4P accumulation at endosomes (Fig. 5 B and Table 1). Certainly, conditions wherein the ER-localized Sac1 phosphatase

is limited allowed us to detect increased PI4P accumulation at the ER compared with conditions where native Sac1 was still active (Fig. 5 C).

Our most consistent finding from all three approaches was that they all failed to identify substantial PM-associated pools of PI (Table 1). Although surprising, this observation is consistent with biochemical quantification from rapidly isolated PM sheets (Saheki et al., 2016). This finding is significant in that it moves the onus for regulating PM PPIn synthesis from the PI4K-catalyzed PI phosphorylation to the supply of PI substrate for this enzyme.

The need for resupply of ER-synthesized PI to the PM for PPIn synthesis has been known for a good many years (Lapetina and Michell, 1973). Vesicular transport of PI4P or PI from the Golgi, an organelle that we show is replete with both lipids, can contribute to PM PI(4,5)P$_2$ synthesis (Szentpetery et al., 2010; Dickson et al., 2014). However, these contributions do not explain the full capacity for PM synthesis, and the speed of the secretory pathway from the ER does not match the rate at which PM pools of PPIn can be turned over; instead, the activity of PI transfer proteins has been proposed to feed PPIn synthesis directly from the ER (Lapetina and Michell, 1973). More recently, specific PI transfer proteins such as Nir2 and TMEM24 have been identified in PM PPIn resynthesis after activation of PLC (Lees et al., 2017a; Dornan et al., 2018). Intriguingly, several of these studies found that simple overexpression of these transfer proteins can accelerate synthesis or elevate steady-state levels of PM PPIn (Chang et al., 2013; Kim et al., 2015; Lees et al., 2017a). Such observations are easily explained when viewed through the prism of our findings, which demonstrate a limited supply of PI at the PM; expanding the supply process is thus expected to increase synthesis or steady-state levels. However, they do not explain how cells control steady-state PI4P and PI(4,5)P$_2$ accumulation. Recently, a regulatory mechanism that puts a break on PI4P catabolism at the PM when levels of PI(4,5)P$_2$ drop was demonstrated (Sohn et al., 2018). However, what homeostatic mechanism(s) maintains steady-state levels of PI(4,5)P$_2$ and how this mechanism couples to the transport of PI to the PM are significant questions for the future.

## Materials and methods
### Plasmids and cloning
Production of new plasmids for this study was accomplished by standard restriction-ligation or NEBuilder HiFi DNA Assembly (New England Biolabs; E5520S), typically using Clontech pEGFP-C1

Figure 6. **Recruitment of PI4K to the PM reveals a scarcity of PI. (A)** Recruitment of PI4KA[N1001] shows no increase in PI4P or PI(4,5)P$_2$. COS-7 cells imaged by TIRFM expressing GFP-P4Mx1 or Tubby$_c$[R332H]-mCherry together with FKBP-PI4K[N1001] and Lyn[N11]-FRB were treated with 1 µM rapamycin at time 0 to induce dimerization. Scale bar = 20 µm. The curves at right show the mean change in reporter intensity, with SEM shaded. The violin plots show AUC analysis of the curves, with the number of cells (pooled across three independent experiments) and the Mann-Whitney $U$ statistic and P value from a two-tailed test. **(B)** PI4KB induces PI4P increases at the mitochondria but not the PM. COS-7 cells were transfected with P4Mx1 to detect PI4P increases by confocal microscopy in conjunction with the indicated compartment-specific FRB and FKBP-PI4KB. Insets are 15 µm and serve as scale bars. Line graphs are means with SEM shaded; inset violin plot shows AUC analysis, with number of cells (pooled across three independent experiments) and the Mann-Whitney $U$ statistic and P value from a two-tailed test. **(C)** PI4KB is active in the PM. COS-7 cells were cotransfected with PI(4,5)P$_2$ biosensor Tubby$_c$-GFP (green) and muscarinic M3

receptors to stimulate PLCβ-induced PI(4,5)P₂ and PI4P depletion from the PM in response to carbachol treatment. Subsequent treatment with the muscarinic antagonist atropine induces resynthesis of PI4P and PI(4,5)P₂ via endogenous PI4KA. Where indicated, the PI4KA inhibitor A1 was added at a 30-nM concentration. Images show confocal sections (scale bar = 20 µm) before stimulation, after addition of carbachol, and after atropine addition as indicated. Curves are means with SEM shaded. The violin plot shows AUC analysis for the post-atropine addition, with number of cells (three independent experiments), Kruskal-Wallis statistic, and P value from a two-tailed test. P values between individual groups shown are derived from a post hoc Dunn's multiple comparison test. CCh, carbachol; mito, mitochondria.

and -N1 backbones or their derivatives. Our standard fluorophores were a human codon–optimized EGFP derived from *Aequorea victoria* GFP containing F64L and S65T mutations (Cormack et al., 1996), a *Discoma* DsRed monomeric variant known as mCherry (Shaner et al., 2004), the iRFP variant derived from *Rhodopseudomonas palustris* bacteriophytochrome BphP2 (Filonov et al., 2011), and mTagBFP2 derived from *Entacmaea quadricolor* protein eqFP578 (Subach et al., 2011). Mutated constructs were generated by site-directed mutagenesis using targeted pairs of DNA oligos. All custom oligos were supplied by Thermo Fisher. After cloning, all constructs were sequence verified by Sanger DNA sequencing. Plasmids constructed for this study are available through Addgene. Table 2 lists all plasmids used in this study and their respective sources.

### Chemicals and reagents
For chemical dimerization experiments, rapamycin (Thermo Fisher; BP2963-1) was dissolved to 1 mM in DMSO as a stock and used at a final concentration of 1 µM in cells. Carbachol (Thermo Fisher; AC10824-0050) was prepared by dissolving in water to 50 mM and stored at –20°C. Atropine (Thermo Fisher; AC226680100) was dissolved to 25 mM in 100% ethanol and stored at –20°C. 30% $H_2O_2$ (EMD Millipore; HX0635-3) was stored at 4°C and prepared fresh in complete imaging media when used as an additive. A1 PI4KA inhibitor (Bojjireddy et al., 2014), a kind gift from Tamas Balla (National Institutes of Health, Bethesda, MD), was dissolved in DMSO to a 100-µM stock and stored at –20°C.

### Cell culture and lipofection
COS-7 (ATCC CRL-1651) cells were cultured at 37°C with a humidified 5% $CO_2$ atmosphere in low glucose DMEM (Thermo Fisher; 10567022) supplemented with 10% heat-inactivated FBS (Thermo Fisher; 10438–034), 100 U/ml penicillin-streptomycin (Thermo Fisher; 15140122), and 0.1% chemically defined lipid supplement (Thermo Fisher; 11905031). Cells were passaged two times a week 1:5 in culture-treated flasks using TrypLE dissociation media (Thermo Fisher; 12604039).

In preparation for imaging, cells were seeded onto 35-mm #1.5 cover glass-bottom dishes (CellVis; D35-20-1.5-N) precoated with 5 µg of human fibronectin (Thermo Fisher; 33016–015) dissolved in water. Once adherent and between 50% and 80% confluent, cells were transfected according to manufacturer's instructions using 1 µg of total DNA complexed with 3 µg of Lipofectamine 2000 (Thermo Fisher; 11668019) in 200 µl Opti-MEM (Thermo Fisher; 51985091). Imaging of transfected samples occurred after an incubation time of ∼18–24 h.

### Fluorescent lipids
TopFluor conjugated PI, PS, and PI(4,5)P₂ were obtained from Avanti Polar Lipids Inc. (810187P, 810283P, and 810184P, respectively). Lipid-BSA complexing was accomplished using 5 µM (0.34 mg/ml) fatty acid–free BSA (Sigma; 126575) dissolved in PBS with 3 mM EGTA. Fluorescent lipids were sonicated in methanol at a concentration of 1 mM before injecting 5 µl into 1 ml BSA/PBS mixture and vortexed at medium to high speed and incubated at 37°C for 20–30 min. Back-extraction media were

Table 1.  **Summary of results**

| Organelle | Is PI detected with the indicated method? | | | Possible reasons for discrepancy |
|---|---|---|---|---|
| | TopFluor-PI | PI-PLC: PI→DAG | PI4K: PI→PI4P | |
| Golgi | Yes | Yes | Yes | No discrepancy |
| Lysosomes | Not tested | Yes | Trace | PI4P phosphatase (e.g., SAC2)? |
| Endosomes | Not tested | Yes | Trace | PI4P phosphatase (e.g., SAC2)? |
| Mitochondria | Yes | Yes[a] | Yes | No discrepancy |
| Peroxisomes | Not tested | Trace | Yes | Failure to target PI-PLC to peroxisomes? |
| ER | Yes | Trace | Yes[b] | Failure to target PI-PLC to ER? |
| PM | Not detected | Trace[c] | Not detected | PI level beneath threshold of detection? |

Results were characterized as "yes" when a substantial and statistically significant signal was observed and "not detected" when a signal was neither significant nor substantial. We used "trace" to denote situations in which statistically significant but small signals ($\Delta F/F_{pre} < 10\%$) were observed. "Not detected" refers to situations in which no accumulation of signal was observed. Note that for TopFluor-PI, not all organelles were interrogated.
[a]Only detected after forced targeting of PI-PLC to the mitochondrial surface.
[b]Only detected after treatment of cells with 500 µM peroxide to inhibit endogenous PI4P phosphatase SAC1.
[c]Only detected transiently after forced targeting of PI-PLC to the PM with a DAG sensor, and inferred by accumulation of DAG-regulated PI/PA transfer protein Nir2.

Table 2. **Plasmids used in this study**

| Plasmid | Vector | Insert | Reference |
|---|---|---|---|
| NES-EGFP-P4Mx1 | pEGFP-C1 | X. leavis map2k1.L(32-44):EGFP:L. pneumophila SidM(546-647) | Sohn et al., 2018 |
| Lyn$^{N11}$-FRB-iRFP | piRFP-N1 | LYN(1-11):MTOR(2021-2113):iRFP | Hammond et al., 2014 |
| LAMP1-FRB-iRFP | piRFP-N1 | LAMP1:MTOR(2021-2113):iRFP | Goulden et al., 2019 |
| EGFP-FYVE-EEA1 | pEGFP-C1 | EGFP:EEA1(1253-1411) | Balla et al., 2000 |
| NES-EGFP-P4Mx1 | pEGFP-C1 | X. leavis map2k1.L(32-44):EGFP:L. pneumophila SidM(546-647) | Zewe et al., 2018 |
| mCherry-Rab5 | pmCherry-C1 | mCherry:Canis lupus RAB5A | Hammond et al., 2014 |
| iRFP-N1-PI-PLC$^{C100}$-FRB | piRFP-N1 | L. monocytogenes PI-PLC(188-287):IGTAGPRSANS[GA]$_4$: MTOR(2021-2113):iRFP | This study |
| TagBFP2-C1-FKB-PI-PLC$^{N187}$ | pTagBFP-C1 | TagBFP2:FKBP1A(3-108):[GGSA]$_4$GG:L. monocytogenes PI-PLC(1-187) | This study |
| pTagBFP2-C1-lynN$^{11}$-FKBP -PI-PLC$^{N187}$ | pTagBFP-C1 | LYN(1-11):RSANS[GA]$_4$:TagBFP2: FKBP1A(3-108): [GGSA]$_4$GG:L. monocytogenes PI-PLC(1-187) | This study |
| pTagBFP2-C1-AKAP1$^{N31}$-FKBP-PI-PLC$^{N187}$ | pTagBFP-C1 | Mus musculus Akap1(1-31) M16L: PTRSANS[GA]$_4$ILSRM:TagBFP2: FKBP1A(3-108): [GGSA]$_4$GG:L. monocytogenes PI-PLC(1-187) | This study |
| mCherry-Rab7 | pmCherry-C1 | mCherry:Canis lupus RAB7A | Hammond et al., 2014 |
| mCherry-FKBP-PI4KB | pmCherry-C1 | mCherry: FKBP1A(3-108): GGSA]$_4$GG:PI4KB | This study |
| mCherry-FKBP-PI4KA$^{C1001}$ | pmCherry-C1 | mCherry: FKBP1A(3-108): GGSA]$_4$GG:PI4KA(1102-2103) | This study |
| mCherry-FKBP-PI4KA$^{C1001-D1957A}$ | pmCherry-C1 | mCherry: FKBP1A(3-108): GGSA]$_4$GG:PI4KA(1102-2103)-Asp1957Ala | This study |
| piRFP-FRB-Giantin | piRFP-C1 | iRFP:MTOR(2021-2113):[GGSA]$_2$:GOLGB1(3097-3226) | This study |
| pmCherry-C1ab-Prkd1 | pmCherry-C1 | mCherry:Mus musculus Prkd1(138-343) | This study |
| NES-GFP-C1ab-Prkd1 | pEGFP-C1 | EGFP:Mus musculus Prkd1(138-343) | Kim et al., 2011 |
| Tubby$_c$-mCherry | pEGFP-N1 | Mus musculus Tub(243-505):mCherry | Quinn et al., 2008 |
| Tubby$_c$$^{R332H}$-mCherry | pEGFP-N1 | Mus musculus Tub(243-505) R332H:mCherry | Quinn et al., 2008 |
| Akap1(31)-FRB-iRFP | piRFP-N1 | Mus musculus Akap1(1-31) M16L:MTOR(2021-2113):iRFP | This study |
| piRFP-FRB-PMP-C-10 | piRFP-C1 | iRFP: MTOR(2021-2113):[GGSA]$_2$QASNSAVSGLRSGSSGG:PXMP(2-195) | This study |
| iRFP-FRB-ER | piRFP-C1 | iRFP:MTOR(2021-2113):[GGSA]$_2$ILNSRV:SACM1L(521-587) | This study |
| β4-GalT$^{N82}$-mCherry | pmCherry-N1 | B4GALT(1-82):mCherry | Addgene #55052[a] |
| iRFP-Sec61β | piRFP-C1 | iRFP:SEC61B | Zewe et al., 2018 |
| COX8A$^{N29}$x2-mCherry | pmCherry-N1 | COX8A(1-29):mCherry | This study |
| MannII$^{N21}$-mKO2 | pmKO2-N1 | Mus musculus Man2a(1-102):Kusabira Orange2: | Addgene #57881[a] |
| LAMP1-mRFP | pmRFP-N1 | LAMP1:mRFP | Jović et al., 2012 |
| mCherry-PMP | pmCherry-C1 | mCherry:SGLRSRAQASNSAV:PXMP(2-195) | |
| mCherry-VAPB | pmCherry-C1 | mCherry:VAPB | Zewe et al., 2018 |
| HAx3-AChR-M3 | pcDNA3.1 | HAx3:CHRM3(2-590) | J. Wes |
| EGFP-Nir2 | EGFP | EGFP:PITPNM1 | Kim et al., 2015 |

[a]β4-GalT$^{N82}$-mCherry (Addgene plasmid #55052; RRID:Addgene_55052) and MannII$^{N21}$-mKO2 (Addgene plasmid #57881; RRID:Addgene_57881) were gifts from Michael Davidson (Florida State University, deceased).

made using 17 mg/ml fatty acid–free BSA (i.e., 250 µM, five times the final concentration applied to cells) in serum-free imaging media and filtered.

## TLC
LK6D 60 Å silica gel 20 × 20–cm glass-backed TLC plates (Whatman; 4865–821) were prepared by dipping into 74 mM sodium oxalate in 0.5% boric acid and dried overnight. Before loading, plates were preequilibrated in a 70:70:4:16 chloroform, methanol, ammonium hydroxide, and water mixture. After loading, the plates were run in the same solvent mixture and then removed and allowed to dry for 15 min at room temperature in a fume hood. Fluorescent lipids were imaged using a ChemiDoc MP imaging system (BioRad) using 460–490-nm LED epi-illumination and a 518–546-nm emission filter.

Cells were extracted using a modified protocol (Lees et al., 1959). Briefly, cells in 35-mm glass-bottom dishes were lysed in 250 µl of 1 M HCl, scraped, and collected in polypropylene tubes. The dishes were rinsed with 333 µl methanol, which was then pooled with the HCl extract. 667 µl chloroform was then added

to the extracts before samples were vortexed vigorously and centrifuged to resolve two phases. The lower phase was removed and washed with 3:48:47 chloroform:methanol:1 M HCl, and the upper phase was reextracted with 86:14:1 chloroform:methanol:1 M HCl. After recentrifuging to resolve phases, the washed lower phase was transferred to a fresh polypropylene tube, and the reextracted upper phase was washed. This washed reextract was pooled with the original extract and dried under nitrogen. Samples were redissolved in 30 µl of an 86:14:1 mixture of chloroform, methanol, and 1 M HCl and streaked into the concentration zone of the TLC plate to run.

### Microscopy

For all live-cell imaging, standard growth medium was replaced with Fluorobrite DMEM (Thermo Fisher; A1896702) supplemented with 10% heat-inactivated FBS, 25 mM Hepes (pH 7.4), 0.1% chemically defined lipid supplement, and 2 mM GlutaMAX (Thermo Fisher; 35050061). The initial volume of imaging media was adjusted so that 2 ml total volume was achieved after all chemical additions for a given experiment.

Confocal imaging was accomplished on a Nikon TiE A1R platform acquiring images in resonant mode with a 100× 1.45 NA plan-apochromatic objective. Signal-to-noise ratio was improved by taking eight frame averages. Excitation of fluorophores was accomplished via a dual fiber–coupled LUN-V laser launch with 405-nm (BFP), 488-nm (GFP), 561-nm (mCherry), and 640-nm (iRFP) lines. Emission was collected on four separate photomultiplier tubes with blue (425–475 nm), green (500–550 nm), yellow/orange (570–620 nm), and far-red (663–737 nm) filters. Blue and yellow/orange channels were recorded concurrently, as were green and far red. Confocal pinhole size was defined as 1.2× the Airy disc size of the longest wavelength channel used in the experiment. Nikon Elements software was used to acquire images for all experiments. All data were saved with the file extension ND2.

An independent Nikon TiE platform coupled with a *TIRF* illuminator arm (Nikon) and 100× 1.45 NA plan-apochromatic objective was used to acquire TIRFM imaging data. Excitation of fluorophores was accomplished via an Oxxius L4C laser launch with 405-nm (BFP), 488-nm (EGFP), 561-nm (mCherry), and 638-nm (iRFP) lines. Emission was collected through dual pass filters from Chroma: blue/yellow-orange (420–480 nm/570–620 nm) and green/far-red (505–550 nm/650–850 nm). A Zyla 5.5 sCMOS camera (Andor) was used to capture images, binning 2 × 2 pixels. Nikon Elements software was used to acquire images for all experiments. All data were saved with the file extension ND2.

### Image analysis

Analysis of images was accomplished in Fiji software (Schindelin et al., 2012). A custom macro was written to generate channel-specific montages and display all x,y positions captured in a given experiment in concatenated series. Individual regions of interest (ROIs) were then generated around cells displayed in these montages.

The ratio of fluorescence intensity between specific compartments in confocal images was analyzed as described

previously (Zewe et al., 2018). A custom macro was used to generate a binary mask through à trous wavelet decomposition (Olivo-Marin, 2002). The mask was applied to measure fluorescence intensity within a given compartment, while normalizing to the ROI's mean pixel intensity to account for variance in expression level present in transient transfections.

To analyze TIRFM images, a minimum intensity projection was used as the basis to generate ROIs within the footprint of individual cells. Background fluorescence values were measured and subtracted from images at all time points. Fluorescence values were then normalized to the ROI's mean pixel intensity of time points preceding treatment or stimulation ($F_{pre}$).

For statistical analysis and generation of graphs and plots, quantitative data were imported into Prism 8 (GraphPad). For area under the curve (AUC) analysis, baseline-corrected data were first sorted into groups of individual curves. The net AUC for each ROI was then pooled and compared among conditions. D'Agostino and Pearson normality tests returned values that significantly varied from a normal distribution, so data were subjected to a nonparametric Kruskal-Wallis test; if significant variance between medians was found, Dunn's multiple comparison test was run post hoc.

All representative images were selected based on a robust signal-to-noise ratio, typical morphology, and fluorescence measurements near the median of their cohort. Any adjustments made to brightness or contrast to ease visibility were made in a linear fashion across the entire image.

### Online supplemental material

Fig. S1 shows controls for cross-talk for the data presented in Fig. 1, C–E. Fig. S2 shows the kinetics of recruitment of split PI-PLC to organelle-specific compartments. Fig. S3 and Fig. S4 show kinetics of recruitment of FKBP-PI4KA and -PI4KB, respectively.

## Acknowledgments

We thank our colleagues cited in the Materials and methods for generously sharing plasmids. We are grateful to our colleague Linton Traub (University of Pittsburgh, Pittsburgh, PA) for assistance with TLC.

This work was supported by National Institutes of Health grant 1R35GM119412-01 (to G.R.V. Hammond).

The authors declare no competing financial interests.

Author contributions: J.P. Zewe, R.C. Wills, B.D. Goulden, and G.R.V. Hammond conceived the experiments and developed methods. J.P. Zewe, A.M. Miller, S. Sangappa, R.C. Wills, and G.R.V. Hammond performed the experiments. J.P. Zewe, A.M. Miller, S. Sangappa, R.C. Wills, and G.R.V. Hammond analyzed the data. G.R.V. Hammond acquired grant funding for this study. G.R.V. Hammond wrote the original draft of the manuscript. All authors reviewed and edited the manuscript.

Submitted: 19 June 2019

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

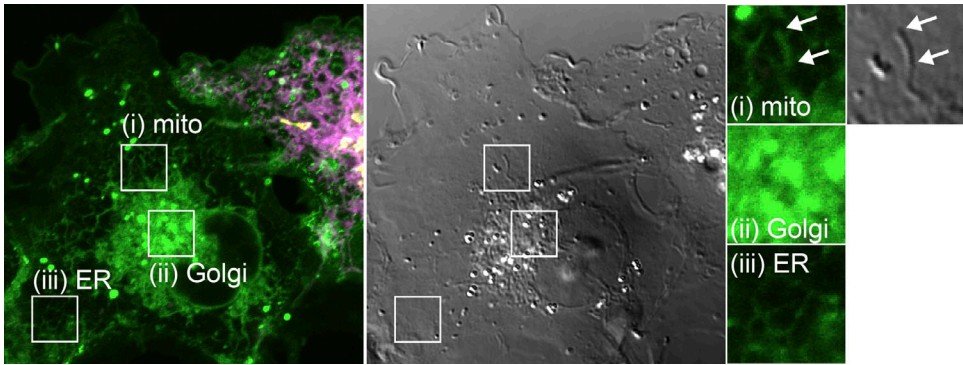

Figure S1. **TopFluor-PI fluorescence distribution is not contaminated by bleed-through from transfected organelle markers**. Images show a non-transfected cell imaged under identical conditions to those presented in Fig. 1. Clear mitochondrial (mito; i), Golgi (ii), and ER (III) morphology of the green fluorescence is seen even with no expression of markers for these compartments, demonstrating that they are not due to fluorescence bleed-through. Note, mitochondrial morphology is evident from the differential interference contrast image (gray). Insets are 7.3 μm and serve as scale bar.

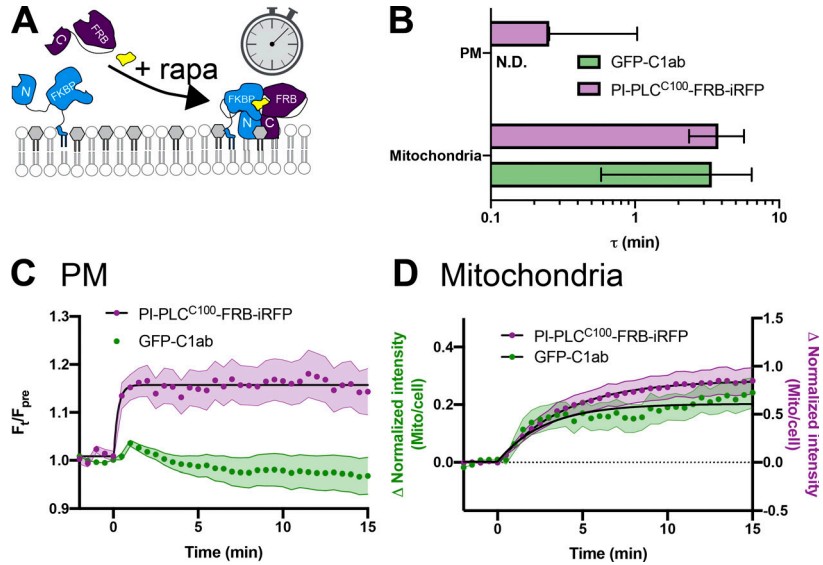

Figure S2. **Kinetics of PI-PLC^C100-FRB-iRFP recruitment to organelle-targeted BFP-FKBP-PI-PLC^N187. (A)** Schematic. **(B)** Summary data. Mean time constant ± 95% confidence interval is shown for each organelle-targeted construct. **(C and D)** Data for C1ab recruitment as shown in C (Fig. 3 A) and D (Fig. 2 C) is shown alongside that for PI-PLC^C100-FRB-iRFP from the same cells. Data are means with SEM shaded; black fits represent the mean fit for all cells to the single-phase exponential ΔIntensity = Plateau × $e^{-(\text{time}/\tau)}$. mito, mitochondria.

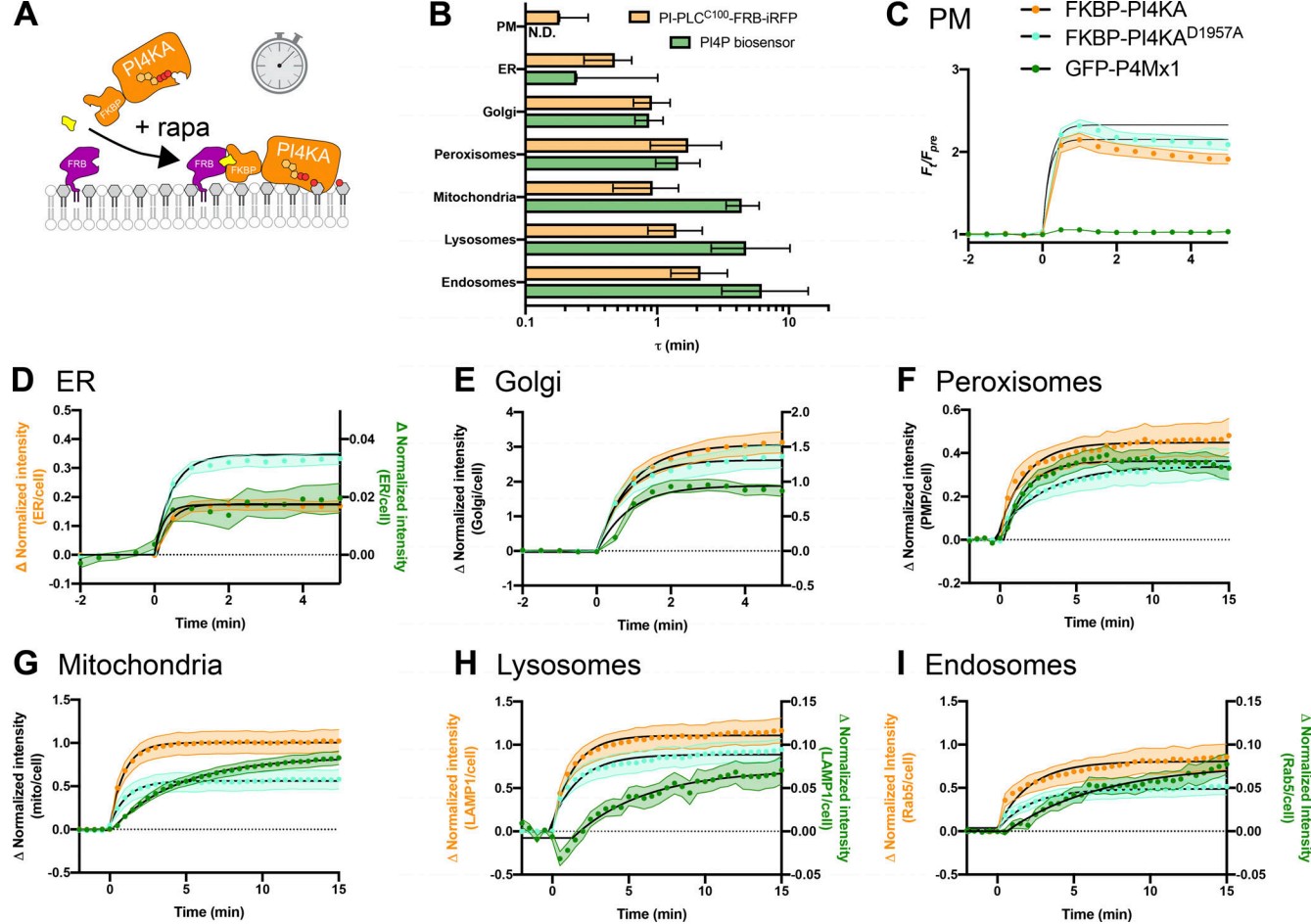

Figure S3. **Kinetics of mCherry-FKBP-PI4KA$^{C1001}$ recruitment to organelle-targeted FRB. (A)** Schematic. **(B)** Summary data. Mean time constant ± 95% confidence interval is shown for each organelle-targeted construct. **(C–I)** Data for PI4P biosensor (green) recruitment as shown in C (Fig. 6 A) and D–I (Fig. 5 B) is shown alongside that for mCherry-FKBP-PI4KA$^{C1001}$ (orange) from the same cells. Data are means with SEM shaded; black fits represent the mean fit for all cells to the single-phase exponential ΔIntensity = Plateau × $e^{-(time/\tau)}$.

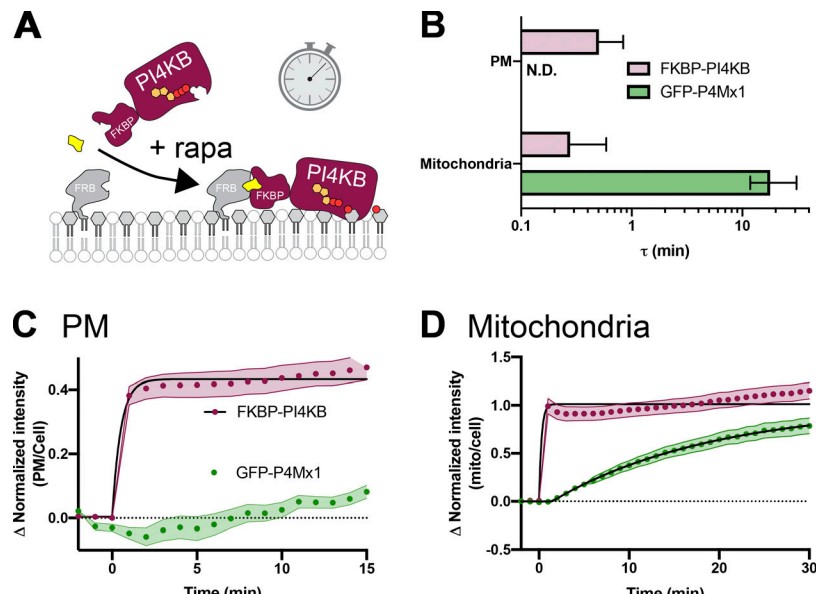

Figure S4.  **Kinetics of mCherry-FKBP-PI4KB recruitment to organelle-targeted FRB. (A)** Schematic. **(B)** Summary data. Mean time constant ± 95% confidence interval is shown for PM (C) and mitochondria (D). Data for GFP-P4Mx1 PI4P biosensor (green) recruitment as shown in Fig. 6 B is shown alongside that for mCherry-FKBP-PI4KB (maroon) from the same cells. Data are means with SEM shaded; black fits represent the mean fit for all cells to the single-phase exponential ΔIntensity = Plateau × $e^{-(\text{time}/\tau)}$. mito, mitochondria.

