## [Peer Review File · The Journal of Cell Biology]

Probing the Subcellular Distribution of Phosphatidylinositol Reveals a Surprising Lack at the PM

James Zewe, April Miller, Sahana Sangappa, Rachel Wills, Brady Goulden, and Gerry Hammond

Corresponding Author(s): Gerry Hammond, Department of Cell Biology, University of Pittsburgh School of Medicine

Review Timeline:

Submission Date:	2019-06-19
Editorial Decision:	2019-07-24
Revision Received:	2019-10-14
Editorial Decision:	2019-10-29
Revision Received:	2019-11-15

Monitoring Editor: Jodi Nunnari

Scientific Editor: Melina Casadio

Transaction Report:

DOI: <https://doi.org/10.1083/jcb.201906127>

July 24, 2019

Re: JCB manuscript #201906127

Dr. Gerry R Hammond
Department of Cell Biology
University of Pittsburgh School of Medicine
BST-South, Room #327 3500 Terrace St
Pittsburgh, PA 15261

Dear Dr. Hammond,

Thank you for submitting your manuscript entitled "Probing the Subcellular Distribution of Phosphatidylinositol Reveals a Surprising Lack at the Plasma Membrane" and thank you for your patience with the review process. The manuscript was assessed by expert reviewers, whose comments are appended to this letter. We invite you to submit a revision if you can address the reviewers' key concerns, as outlined here.

Given the focus of the work on developing approaches to visualize phosphatidylinositol (PI) distribution in cells, we asked the referees to provide comments on the paper as a "Tools" submission (a format aimed at presenting new methods or datasets of immediate value and broad utility to the cell biology community, with new cell biological insight provided to demonstrate the value of the tool.) You will see that, although the referees shared different levels of enthusiasm, they found the experimental design elegant and the discovery that PI is poorly detectable at the plasma membrane interesting and exciting.

The reviewers suggested experiments to strengthen these results and validate the methods. In our view, the primary focus of the revision should be on addressing the questions raised by Reviewers #2 and #3 about the split tool regarding its activity in different contexts. Reviewer #1's request to add a table/graphic comparing the three approaches for their ability to visualize PI per compartment is also an excellent idea. All of Reviewers #2 and #3's additional points are reasonable and relevant to your core conclusions and should be addressed as well.

While you are revising your manuscript, please also attend to the following editorial points to help expedite the publication of your manuscript. Please direct any editorial questions to the journal office. **Please also submit the revision in the "Tools" format - please let us know if you need any help with this change at resubmission. There is no difference in manuscript organization and formatting between the Tools and Article formats, so no change is needed in that regard.**

GENERAL GUIDELINES:

Text limits: Character count for Tools is < 40,000, not including spaces. Count includes title page, abstract, introduction, results, discussion, acknowledgments, and figure legends. Count does not include materials and methods, references, tables, or supplemental legends.

Figures: Tools may have up to 10 main text figures. Figures must be prepared according to the policies outlined in our Instructions to Authors, under Data Presentation,

<http://jcb.rupress.org/site/misc/ifora.xhtml>. All figures in accepted manuscripts will be screened prior to publication.

Supplemental information: There are strict limits on the allowable amount of supplemental data. Tools may have up to 5 supplemental figures. Up to 10 supplemental videos or flash animations are allowed. A summary of all supplemental material should appear at the end of the Materials and methods section.

The typical timeframe for revisions is three months; if submitted within this timeframe, novelty will not be reassessed at the final decision. Please note that papers are generally considered through only one revision cycle, so any revised manuscript will likely be either accepted or rejected.

Thank you for this interesting contribution to the Journal of Cell Biology. You can contact us at the journal office with any questions, cellbio@rockefeller.edu or call (212) 327-8588.

Sincerely,

Jodi Nunnari, Ph.D.
Editor-in-Chief, Journal of Cell Biology

Melina Casadio, Ph.D.
Senior Scientific Editor, Journal of Cell Biology

Reviewer #1 (Comments to the Authors (Required)):

This study uses three complimentary approaches to investigate the intracellular distribution of PI. The work is well done and this careful comparison of the three approaches will be useful for the field. The most notable conclusion is that there is a relatively small amount of PI in the PM. This is an important finding since, as the authors point out, it suggests that PI transport to the PM plays an important role in PM phosphoinositide metabolism. However, the conclusions from the rest of the study are somewhat disappointing. They show, not surprisingly, that PI is not visualized equally well by all the approaches. What accounts for these differences is not resolved. The study would be more useful for a broad audience if it included a table or chart comparing the ability of the three approaches to visualize PI in each of the organelles tested.

Reviewer #2 (Comments to the Authors (Required)):

The work of Zewe et al., uses a combination of molecular and imaging approaches in an attempt to characterize the cellular distribution of phosphatidylinositol, an essential lipid that serves as a precursor for other phosphoinositide species. Although much of their data is indirect, the authors have elegantly designed experiments to leverage the maximum amount of information, while also noting in detail the limitations of experiments and interpretation of data. From their indirect evidence the authors determine that most organelles have some resting PI, with the surprising exception of the plasma membrane. Below I have some specific comments that may help improve the work

Major Comments:

1. Given the fluorescent label on the exogenously applied TopFluor-PI is present on the cytoplasmic tails, how can the authors be sure that they are monitoring TopFluor-PI and not the metabolites of TopFluor (e.g. TopFluor-DAG) once the labeled lipid enters the cell? Further, the description of the TLC experiments in Fig.1B is not clear. Is the added standard PtdIn or DAG? It may be important to demonstrate that a DAG standard can be detected, yet in cell extracts little metabolite accumulation occurs.

2. Along similar lines at comment 1. The TopFluor-PS and -PI have different fatty acid chains which could alter the rate of incorporation into the plasma membrane and its extraction from the membrane. The authors may consider experiments with TopFluor-PI of different tail lengths to ensure similar results.

3. Reviewer curiosity: Can the authors extract any kinetic information from their TopFluor-PI experiments? Namely, the rate it takes to reach equilibrium in the ER, or Golgi, or mito? Given the relatively short loading time, it is surprising that TopFluor-PI finds its home in many organelle membranes. The rate of accumulation of TopFluor-PI in different membranes may provide evidence for a preferred transport route.

4. What is the affinity of C1ab for DAG? This becomes important when the authors begin making statements like: (Page 12, line 2-3) "This result would be consistent with the notion that in the PM, levels of PI are much lower compared to PI(4,5)P2". This assumes similar affinities of biosensors for lipid substrate.

5. The author experiments with Lyn11-FKBP-PI-PLC are difficult to interpret. How do they know the enzyme has been presented at an orientation to the plasma membrane that permits hydrolysis? Certainly, the change in C1ab is more impressive with receptor activation, yet that simple provides evidence that the DAG sensor is not saturated, it does not confirm the PI-PLC as appropriate activity. Important controls are missing to ensure the formation of C1ab puncta is biologically important. Similar questions concern Figure 4 with the tubby plasmid. Could the authors monitor PM PtdIns4P using their high-affinity P4M probe to determine if PtdIns4P levels at the PM, similar to the Golgi (Figure 4), decrease following PI-PLC recruitment. This would provide an indirect estimate of PI-PLC activity.

6. Figure 6. The authors recruit PI4KB to the plasma membrane and see minor increases in PtdIns4P. Have the authors considered that the catalytic activity of the PI4KB enzyme may be dependent on the acyl-chain composition of the PI at the PM? In other words, for the enzyme to catalyze the reaction, perhaps PI 36:2 has to be present, whereas PI 38:4 may accumulate at the

PM. In such a hypothetical situation the enzyme would have little effect upon recruitment. Given that we know little about the lipid-chain 'code' for phosphoinositides, this is certainly something the authors may consider as a discussion point.

7. To complement other recruitment experiments the authors should recruit the PI-PLC enzyme to determine if the rate of P4M recovery is altered. The reviewer is seeking to find additional evidence that the enzymes is functionally active at the PM.

Minor comments:

1. Figure 1. Control experiments for C and E should be performed to ensure the weak top-fluor signal does not represent bleed-through.

2. Figure 1. Quantification of images should be included to determine the steady-state ratio of TopFluor-PI between organelle compartments.

3. Page 7, line 15 "This implies that PI is widely distributed in the cell", such a comment is not supported by the data. The authors data suggests that exogenously applied TopFluor-PI may partition into different membrane compartments.

4. The overall presentation of data is of a very high standard. One small suggestion is that given the complexity of experiments and multiple fluorescent proteins, all fluorescent proteins be noted in the figure panels.

Reviewer #3 (Comments to the Authors (Required)):

In their manuscript, Hammond's team presents new strategies to detect inside eukaryotic cells phosphatidylinositol (PI), one of the main building-blocks of membrane lipid matrix and the precursor of phosphoinositides. In the recent years, many efforts have been done to follow in living cells by microcopy the metabolism and movement of different lipid species (e.g., phosphatidylserine , phosphatidic acid, diacylglycerol, phosphoinositide, sterol, sphingolipids) but there was no similar approach dedicated to PI. Yet this was quite necessary, primarily to better understand the genesis of PI4P, PI3P or PI(4,5)P2 on the cytosolic side of organelle, these lipids being key molecular players in many cell processes like vesicular trafficking, lipid transport, cytoskeleton dynamics. Here are described three elegant ways to see by confocal and TIRF microscopy the allocation of PI throughout the cell (i) by adding exogenously fluorescent-PI, (ii) by using a split phospholipase whose ability to convert PI into DAG can be triggered on demand to reveal, via the detection of DAG, where are the PI pools amongst organelles or (iii) by measuring the PI-to-PI4P conversion induced by kinases that are recruited onto specific organelles. The mapping of PI levels in the cells requires the use of these different approaches as each one has its own drawback. Summarizing, the authors observe PI in the major organelles (ER, Golgi, mitochondria, peroxysome, endosomes) yet with the unexpected and novel finding that PI is poorly detectable at the plasma membrane, at least in its cytosolic side. This opens new assumptions on the regulation of phosphoinositide synthesis, which maybe more occurs through a regulation of PI supply to the PM than via a direct regulation of PI-kinases. From a technical standpoint, one can consider that the tools can be implemented in any laboratory that owns good microscopes. The manuscript will be of high interest for the audience of JCB ; nevertheless authors should clarify a few points notably if one of the main goal is to provide confident tools for cell biologists.

Major comments

1 - To my opinion the most innovative tool is the split PI-PLC which exists in two flavors (targetable or non-targetable) and is likely the easiest to implement. Data shown in the Figure 2B describe the use of the non-targetable version. The results are convincing at the first glance but then when considering two points they are maybe far more difficult to interpret than experiments in which the phospholipase is targeted to one particular organelle and therefore locally produces DAG. Indeed, because the split PI-PLC is homogeneously distributed throughout the cytosol (Fig 2A), upon adding rapamycin, it should hydrolyze any source of accessible PI. Thus the pictures shown in Figure 2B represent the allocation of the DAG biosensor between the different organelles according to the relative amount of DAG in each organelle. 1) Does the C1 probe recognize DAG efficiently at the surface of each type of organelle, regardless on their specific lipid composition and membrane curvature ? 2) Is the split PI-PLC able to associate with different compartments and hydrolyse DAG with the same efficiency (the authors suggest that the PI-PLC does not necessarily work on the ER or mitochondria) Jointly, the differential ability to generate and/or recognize DAG might strongly bias the output and analysis of the experiment. For instance, the apparent lack of response at the ER surface might arise from a more efficient hydrolysis of PI and acute detection of DAG in other organelles like the Golgi. The authors should document these two points, maybe by biochemical means and via cell fractionation, to more characterize the enzymatic activity of the PLC at interface and the readout of the experiment (i.e the sensibility of the DAG biosensor to membrane context). Such an analysis will serve equally to better compare the recruitable FKBP-PI-PLC construct with endogeneous phospholipases (as in Figure 3). The rigidity (saturated lipids, sterol) or other features (high PS-density) of the PM might strongly impede the enzymatic ability of the exogenous construct but likely not the PLCbeta.

2- A main information is missing in the manuscript and should be provided: the kinetic of recruitment of the targetable split PI-PLC constructs and of other FKBP-tagged lipid-modifying enzymes onto organelles. Is the recruitment of the enzymes is full and immediate after a few seconds or a few minutes? This is necessary to more precisely interpret the kinetics of lipid conversion that theoretically should depend on the level of substrate and amount of enzyme recruited on organelle surface.

Minor comments

P3. One sentence is problematic « Therefore, it is currently unclear how much PI is available for PI synthesis in .." »

Figure 1. The name of the markers for the Golgi, ER and mitochondria should be indicated in the panels C, D and E (as in others Figures)

Figure 2 and other Figures. The injection of rapamycin should be indicated by an arrow in the plots of panel B and in other Figure showing kinetics of lipid conversion after rapamycin treatment.

Figure 2C . Authors should indicate in the Figure to what correspond the green, blue and red channel (iRFP ?) . The cartoon should indicate that the PLCC100-FRBP is tagged with the iRFP

Figure 6 - The recruitment of PI4KA or PI4KB onto the PM does not trigger any production of PI4P or PIP2, suggesting that no substantial PI pool is present in this membrane. However, considering the initial high level of staining of the PM by the GFP-P4M or mCherry-Tubby constructs (pictures in panel A) , prior to rapamycine addition, is there enough phosphoinositide sensor remaining in the cytosolic to follow any additional synthesis of PI4P or PIP2 ?

It is difficult to understand the experiments shown in Figure 6C mainly because the cartoons are a little bit unclear as well as the curves shown on the right (rapamycin is in brown but refer to the blue curve and PI4KB recruitment, the brown curve corresponds to the reference experiment) . Authors must choose between the labeling in brown (rapamycin) or blue (PI4KB). Also, what is the exact order of injection for rapamycin,atropine, carbachol and A1 ?

Dear Jodi and Melina,

We are resubmitting our revised manuscript "Probing the Subcellular Distribution of Phosphatidylinositol Reveals a Surprising Lack at the Plasma Membrane" (manuscript #201906127) for consideration as a *Tools* article in JCB. Our detailed response is outlined below. We are also aware that the Balla lab is resubmitting a manuscript on a similar topic; we would be grateful if the journal could delay consideration or a final decision on our paper until such a decision is reached for Balla's in parallel.

With my best,

Gerry

Editor's/Reviewers' comments are quoted in italics

Our responses appear in-line in blue

Excerpts from the amended manuscript are in red both in this rebuttal and in the revised manuscript.

Editor's Summary:

Given the focus of the work on developing approaches to visualize phosphatidylinositol (PI) distribution in cells, we asked the referees to provide comments on the paper as a "Tools" submission (a format aimed at presenting new methods or datasets of immediate value and broad utility to the cell biology community, with new cell biological insight provided to demonstrate the value of the tool.) You will see that, although the referees shared different levels of enthusiasm, they found the experimental design elegant and the discovery that PI is poorly detectable at the plasma membrane interesting and exciting.

The reviewers suggested experiments to strengthen these results and validate the methods. In our view, the primary focus of the revision should be on addressing the questions raised by Reviewers #2 and #3 about the split tool regarding its activity in different contexts. Reviewer #1's request to add a table/graphic comparing the three approaches for their ability to visualize PI per compartment is also an excellent idea. All of Reviewers #2 and #3's additional points are reasonable and relevant to your core conclusions and should be addressed as well.

We appreciate the careful and constructive comments from both the editor and reviewers, and for the opportunity to respond to them. We detail below a comprehensive and point-by-point response to each reviewer's comments. Notably, the revised manuscript contains substantial new experimental data in **figure 3B** that provides additional evidence for the activity of our split-PI-PLC enzyme at the plasma membrane, as requested by reviewers #2 and #3. We have also included the table suggested by reviewer #1, and several supplementary figures detailing the kinetics of FKBP-tagged enzyme recruitment requested by reviewer #3.

We believe these revisions fully address the reviewers' concerns, and hope they and the editors agree the manuscript is now ready for publication.

Reviewer #1

We thank the reviewer for their constructive comments and are pleased they appreciate the importance of our central conclusions.

Point 1.1: *The study would be more useful for a broad audience if it included a table or chart comparing the ability of the three approaches to visualize PI in each of the organelles tested.*

This is an excellent suggestion that we agree will aid in the reader’s interpretation of our results, whilst also enabling the casual reader to quickly digest our main findings. We are grateful to the reviewer (and editor) for suggesting its inclusion. We now refer to this summary table in the discussion, but we have included it here below for convenience:

Organelle	Is PI detected with the indicated method?			Possible reasons for discrepancy
	TopFluor-PI	PI-PLC: PI→DAG	PI4K: PI→PI4P	
Golgi	Yes	Yes	Yes	No discrepancy
Lysosomes	Not tested	Yes	Trace	PI4P phosphatase e.g. SAC2?
Endosomes	Not tested	Yes	Trace	PI4P phosphatase e.g. SAC2?
Mitochondria	Yes	Yes*	Yes	No discrepancy
Peroxisomes	Not tested	Trace	Yes	Failure to target PI-PLC to peroxisomes?
ER	Yes	Trace	Yes [#]	Failure to target PI-PLC to ER?
PM	Not detected	Trace [‡]	Not detected	PI level beneath threshold of detection?

Table 1: summary of results. Results were characterized as “yes” when substantial and statistically significant signal was observed and “not detected” when signal was neither significant nor substantial. We use “trace” to denote situations where statistically significant but small signals ($\Delta F/F_{pre} < 10\%$) were observed. “Not detected” refers to situations where no accumulation of signal was observed. Note that for TopFluor-PI, not all organelles were interrogated. *only detected after forced targeting of PI-PLC to the mitochondrial surface. [#]only detected after treatment of cells with 500 μ M peroxide to inhibit endogenous PI4P phosphatase SAC1. [‡]Only detected transiently after forced targeting of PI-PLC to the PM with a DAG sensor, and inferred by accumulation of DAG-regulated PI/PA transfer protein Nir2.

Reviewer #2

We truly appreciate the reviewer’s constructive and insightful suggestions to strengthen our work. We hope that they find our responses comprehensive and conclusive.

Major Comments:

Point 2.1. *Given the fluorescent label on the exogenously applied TopFluor-PI is present on the cytoplasmic tails, how can the authors be sure that they are monitoring TopFluor-PI and not the metabolites of TopFluor (e.g. TopFluor-DAG) once the labeled lipid enters the cell? Further, the description of the TLC experiments in Fig.1B is not clear. Is the added standard PtdIn or DAG? It may be important to demonstrate that a DAG standard can be detected, yet in cell extracts little metabolite accumulation occurs.*

We agree with the reviewer that it is essential to show that the acyl-chain TopFluor-conjugated lipid is not metabolized, especially into DAG. This was precisely why we included the thin-layer chromatography experiments, though we apologize for the lack of clarity in the description of the data. We have now amended the text of the results to more clearly indicate the comparison to a TopFluor-PI standard, and included new data demonstrating that this standard is indeed resolved from a DAG standard. **These data are included in a revised figure 1B and in the results, p7 lines 5-9,** reproduced here for the reviewer's convenience:

“...we extracted lipids from our COS-7 cells loaded with TopFluor-PI after back extraction for analysis by thin-layer chromatography: we observed no such metabolism of PI, **which migrated identically to the TopFluor-PI standard and was well resolved from TopFluor-DAG (figure 1B).** Thus, COS-7 cells appeared to traffic exogenous fluorescent PI molecules from the PM and distribute it to intracellular membranes intact.”

Point 2.2. Along similar lines at comment 1. The TopFluor-PS and -PI have different fatty acid chains which could alter the rate of incorporation into the plasma membrane and its extraction from the membrane. The authors may consider experiments with TopFluor-PI of different tail lengths to ensure similar results.

This is a valid point but unfortunately, we are restricted to the commercially available lipids, which are only available with the current fatty acid tails – so we are unable to perform this experiment.

Whilst we agree with the reviewer that the tails could potentially impact the trafficking fate of the exogenous lipid, both lipids do contain physiologically common fatty acids in the *sn*-1 position (18:1 for PI and 16:0 for PS). These are unlikely to alter trafficking to the extent that the bulky fatty fluorophore in *sn*-2 will. Note, although these fluors are different between PI and PS, in this case the difference is subtle, with the inclusion of a single amide group in the PPI derivatives, which does not change the overall length of the fatty fluor.

Point 2.3. Reviewer curiosity: Can the authors extract any kinetic information from their TopFluor-PI experiments? Namely, the rate it takes to reach equilibrium in the ER, or Golgi, or mito? Given the relatively short loading time, it is surprising that TopFluor-PI finds its home in many organelle membranes. The rate of accumulation of TopFluor-PI in different membranes may provide evidence for a preferred transport route.

We attempted this type of analysis when we first conducted the experiments. However, the intense labelling of the outer leaflet of the plasma membrane before back extraction with excess BSA tends to occlude the intracellular signal (see the images in figure 1A). We have included

data from a representative experiment for the reviewer's interest (in fact, this is the experiment which included the cells in figure 1A). As can be seen, total fluorescence intensity increases by 27.9 ± 0.8 -fold during loading (mean \pm s.e.), and declines to 6.6 ± 0.3 after back extraction. The intense labelling of the PM occludes kinetic analysis of the accumulation in the loading phase, since it is more than 4-fold more intense than the intracellular fluorescence.

Point 2.4. *What is the affinity of C1ab for DAG? This becomes important when the authors begin making statements like: (Page 12, line 2-3) "This result would be consistent with the notion that in the PM, levels of PI are much lower compared to PI(4,5)P2". This assumes similar affinities of biosensors for lipid substrate.*

The affinity of the C1ab domain from PKD1 for DAG is high; for dioleoyl-glycerol it has been estimated at ~ 200 nM (doi: 1042/BJ20071334). Notably, when comparing DAG accumulation in the experiments that are discussed on p12, we are comparing DAG derived from PI using inducible PI-PLC with that derived from PI(4,5)P₂ via activity of endogenous PLC β isoforms. In this case, the relative accumulation of the C1ab sensor (beneath a saturating level) should report relative mass of DAG, and the affinity will not be affected by the source of that DAG. We have clarified the results on **p12, lines 14-16**:

This result would be consistent with the notion that in the PM, levels of PI are much lower compared to PI(4,5)P₂; hydrolysis of a relatively small fraction of PI(4,5)P₂ induced by ATP still produces a much greater increase in DAG than PI-PLC-mediated hydrolysis of PM PI.

Point 2.5a. *The author experiments with Lyn11-FKBP-PI-PLC are difficult to interpret. How do they know the enzyme has been presented at an orientation to the plasma membrane that permits hydrolysis? Certainly, the change in C1ab is more impressive with receptor activation, yet that simply provides evidence that the DAG sensor is not saturated, it does not confirm the PI-PLC as appropriate activity. Important controls are missing to ensure the formation of C1ab puncta is biologically important.*

This is indeed a weakness of the manuscript, which we now acknowledge and mitigate with a new experiment described on **p 12, paragraph beginning line 17** with the data shown in new **figure 3B**. In short, we used translocation of Nir2 to ER-PM junctions to demonstrate that Lyn₁₁-targeted split PI-PLC indeed generates DAG at the PM:

An alternative explanation to these data could be that the Lyn amino-terminal fusion does not orient the split-PI-PLC in an orientation conducive to activity in the PM; the few DAG puncta that we observe may instead be produced on organelle membranes that happen to approach close enough to the PM for the re-combined PI-PLC to hydrolyze their PI. Therefore, in order to test whether PM-targeted PI-PLC indeed generated DAG at the PM, we sought an alternative route to detect PM-localized DAG. To this end, we took advantage of the fact that translocation of the ER-localized PI/phosphatidic acid transfer protein Nir2 requires PM DAG to translocate to ER-PM contact sites (Kim et al., 2015). As shown in figure 3B, dimerization of split PI-PLC at the

PM induced translocation of Nir2 to puncta as seen in TIRFM; this occurred to virtually the same extent as with activation of endogenous PLC β with ATP. Although there were small differences in the kinetics of the translocation, no significant difference between the two stimuli was detected (figure 3B). Together with our observations with the C1ab DAG sensor, these data imply that sufficient PI is present at the PM that, when converted to DAG, can recruit Nir2; but it is far less than the quantity produced when hydrolysis of PI(4,5)P₂ is activated by endogenous PLC β .

(B) Split PI-PLC induces translocation of Nir2 to ER-PM contact sites. Cells were transfected with PM-targeted split PI-PLC and stimulated with rapamycin or ATP as in (A). Translocation of GFP-Nir2 was recorded; images show representative TIRFM images with fluorescence normalized to pre-stimulus levels (F/F_{pre}). Scale bar = 10 μ m. The line graphs show mean F/F_{pre} with s.e.m. shaded; the violin plots show area under the curve analysis of the line graphs with the number of cells from 3 independent experiments indicated, along with results of Wilcoxon signed rank test comparing each population to a hypothesized AUC of 0, as well as a Mann-Whitney U test comparing the differences between AUC after ATP or rapamycin stimulation.

Point 2.5b. Similar questions concern Figure 4 with the tubby plasmid. Could the authors monitor PM PtdIns4P using their high-affinity P4M probe to determine if PtdIns4P levels at the PM, similar to the Golgi (Figure 4), decrease following PI-PLC recruitment. This would provide an indirect estimate of PI-PLC activity.

Although we believe the Nir2 experiments described in response to point 2.5a address the issue of PI-PLC activity at the PM, we have added the requested experiment, as this was a notable absence from the original manuscript. The results and our interpretation are now included in the results beginning p. 13 line 19 onwards and in a new figure 4B:

Unexpectedly, we observed a small (~10%) but significant increase in PM PI4P biosensor, P4M (figure 4B). It is possible that this results from a paradoxical increased supply of PI to the PM, caused by the PI-PLC-induced translocation of Nir2 that we reported in figure 3B. This would require the endogenous PI4K to consume Nir2-delivered PI before the PM-targeted split PI-PLC, which would also be consistent with the failure of PI-PLC to deplete PI4P or PI(4,5)P₂ in these experiments.

Figure 4: PI-PLC leads to little or no depletion of PPI in Golgi and endosomes (A) or PM (B). Cells were transfected with PI-PLC¹⁰⁰-FRB, FKBP-PI-PLC^{N100} (or PM-targeted Lyn^{N11}-FKBP-PI-PLC^{N100}) and GFP-PKD1-C1ab to detect DAG, along with the indicated PPI biosensor. Dimerization and activation of PI-PLC was induced with 1 μ M rapamycin as indicated. Images are confocal sections (A) or TIRFM (B). Scale bars = 20 μ m. Line graphs show the change in compartment specific fluorescence of C1ab (green) or the PPI biosensor (magenta) normalized to pre-rapamycin levels (F_{pre}). The violin plots show area under the curve, with the number of cells (pooled across 3 independent experiments), the sum of signed ranks (W) and P-value from a two-tailed Wilcoxon signed rank test comparing to a null hypothesis area under the curve value of 0 (with a baseline of 1).

Point 2.6. Figure 6. The authors recruit PI4KB to the plasma membrane and see minor increases in PtdIns4P. Have the authors considered that the catalytic activity of the PI4KB enzyme may be dependent on the acyl-chain composition of the PI at the PM? In other words, for the enzyme to catalyze the reaction, perhaps PI 36:2 has to be present, whereas PI 38:4 may accumulate at the PM. In such a hypothetical situation the enzyme would have little effect upon recruitment. Given that we know little about the lipid-chain 'code' for phosphoinositides, this is certainly something the authors may consider as a discussion point.

We agree with the reviewer that it is essential to demonstrate catalytic competence of the PI4KB enzyme at the PM, which is why we included the experiments shown in figure 6C. However, we were remiss to include the possibility raised by the reviewer, that PI4KB is incompatible with PM PI acylation status. We have amended this in the results on **p. 18, lines 18-29**:

We next sought to demonstrate that PI4KB can intrinsically be active at the PM, ruling out biological insufficiency of PI4KB or that the enzyme has a preference for acyl chains found in Golgi but not PM PI.

Point 2.7. To complement other recruitment experiments the authors should recruit the PI-PLC enzyme to determine if the rate of P4M recovery is altered. The reviewer is seeking to find additional evidence that the enzymes is functionally active at the PM.

We indeed performed this experiment in the course of our studies, and we include the results here. In short, we did not observe inhibition of PI(4,5)P₂ re-synthesis after inactivating muscarinic receptors. However, we do not believe this experiment differentiates the reviewer's legitimate hypothesis (that split PI-PLC is not active at the PM) from our explanation given in point 2.5b: that PI-PLC does not impact steady-state PM PPI_n levels because there is little PI present in the PM, and the newly delivered PI molecules are consumed efficiently by PM PI4KA (and much more efficiently than by PI-PLC). Therefore, we do not think these data add to central conclusions of the paper and prefer not to include them.

Minor comments:

Point 2.8. Figure 1. Control experiments for C and E should be performed to ensure the weak top-fluor signal does not represent bleed-through.

We have now included new **supplementary figure 1** showing an appropriate control to eliminate bleed-through. Also note, in the “microscopy” section of the **methods on pp. 24-25**, we detail how we configure our imaging in both confocal and TIRFM to acquire spectrally overlapping fluorescence channels independently to eliminate excitation and emission cross-talk. However, the reviewer is quite correct to demand this experiment to eliminate red-shifted fluorescence caused by excimers of the fluorophore at high concentration, which is known to occur with the related BODIPY dye.

Figure S1: TopFluor-PI fluorescence distribution is not contaminated by bleed-through from transfected organelle markers. Images show a non-transfected cell imaged under identical conditions to those presented in figure 1. Clear mitochondrial (i), Golgi (ii) and ER (iii) morphology of the green fluorescence is seen even with no expression of markers for these compartments, demonstrating that they are not due to fluorescence bleed-through. N.B. mitochondrial morphology is evident from the differential interference contrast image (gray).

Point 2.9. Figure 1. Quantification of images should be included to determine the steady-state ratio of TopFluor-PI between organelle compartments.

We did not perform a quantitative analysis of co-localization because this would not produce an accurate steady-state estimate of TopFluor-PI distribution. This is because ER, Golgi and mitochondria are not resolved from one another in the juxta-nuclear region. For example, in figure 1 E, morphology of the green fluorescence that matches with the mitochondrial marker can be seen, but it is contiguous with (i.e., not resolved from) underlying ER. Thus any measurement of steady-state distribution of fluorescence incorporates fluorescence from both mitochondria and ER.

Note that this situation differs from our subsequent experiments measuring DAG and PI4P biosensors since here, the change in fluorescence intensity at a particular organelle relative to the rest of the cell is considered. Thus we do not estimate the steady-state – only the change in steady state.

Point 2.10: *Page 7, line 15 "This implies that PI is widely distributed in the cell", such a comment is not supported by the data. The authors data suggests that exogenously applied TopFluor-PI may partition into different membrane compartments.*

The reviewer is correct, though we stand by the term “implies”, which we selected because it conveys the uncertainty and caveats mentioned later in the same paragraph. We hope the reviewer agrees that our modified statement on **p 7, line 15** is more justifiable and precise:

To the extent that TopFluor-PI traffic and steady-state distribution mirrors natural PI, this implies that PI is widely distributed in the cell...

Point 2.11: *The overall presentation of data is of a very high standard. One small suggestion is that given the complexity of experiments and multiple fluorescent proteins, all fluorescent proteins be noted in the figure panels.*

Although we have comprehensively listed the constructs and fluorescent tags in the methods section, we deliberately omitted fluorescent proteins from the figure panels. This is because we select false colors for clarity and consistency with the illustrations and other data panels in the figure, and in an attempt to accommodate readers with color vision deficiencies (for example, our use of magenta instead of red when illustrating co-localization with green). We are concerned that including a fluorescent protein label in the panels with an incongruent false color could induce the “Stroop effect” in the reader. Therefore, we respectfully decline this request.

Reviewer #3

We appreciate the reviewer’s deeply thoughtful and reasonable comments on the manuscript. We hope they agree that the following responses satisfy the queries and strengthen the conclusions.

Major comments

Point 3.1: *To my opinion the most innovative tool is the split PI-PLC which exists in two flavors (targetable or non-targetable) and is likely the easiest to implement. Data shown in the Figure 2B describe the use of the non-targetable version. The results are convincing at the first glance but then when considering two points they are maybe far more difficult to interpret than*

experiments in which the phospholipase is targeted to one particular organelle and therefore locally produces DAG. Indeed, because the split PI-PLC is homogeneously distributed throughout the cytosol (Fig 2A), upon adding rapamycin, it should hydrolyze any source of accessible PI. Thus the pictures shown in Figure 2B represent the allocation of the DAG biosensor between the different organelles according to the relative amount of DAG in each organelle.

The reviewer has raised a number of concerns with respect to the experiments with the split PI-PLC approach, which we will respond to individually.

Point 3.1a: *1) Does the C1 probe recognize DAG efficiently at the surface of each type of organelle, regardless on their specific lipid composition and membrane curvature ?*

Between the data presented in figures 2-3, we can show robust recruitment of the C1ab domain to all membrane compartments except the ER and Golgi. We now specifically elude to a failure to detect DAG in an updated sentence in the results on **p 10, line 18**:

However, it is also possible that the cytosolic enzyme is not active on these membranes, or that the C1ab probe does not efficiently recognize DAG in these membrane contexts.

We also note to the reviewer that whereas we failed to demonstrate substantial C1 recruitment to the ER or peroxisomal membrane, the Balla lab has reported success in this regard in a pre-print (doi: 10.1101/677229) that we believe is being considered beside our own manuscript. Therefore, we believe that this concern will be further alleviated by those findings.

Point 3.1b: *2) Is the split PI-PLC able to associate with different compartments and hydrolyse DAG with the same efficiency (the authors suggest that the PI-PLC does not necessarily work on the ER or mitochondria) Jointly, the differential ability to generate and/or recognize DAG might strongly bias the output and analysis of the experiment. For instance, the apparent lack of response at the ER surface might arise from a more efficient hydrolysis of PI and acute detection of DAG in other organelles like the Golgi.*

These are indeed valid caveats that limit the interpretations of our experiments that were already discussed on p. 10 when we introduced the targetable constructs. However, we have expanded this discussion to include the reviewer's additional possibility on **p 10, line 20**:

Moreover, substantial recruitment of C1ab to DAG-replete membranes like the Golgi may also deplete the probe available to detect smaller pools elsewhere.

Point 3.1c: *The authors should document these two points, maybe by biochemical means and via cell fractionation, to more characterize the enzymatic activity of the PLC at interface and the readout of the experiment (i.e the sensibility of the DAG biosensor to membrane context). Such an analysis will serve equally to better compare the recruitable FKBP-PI-PLC construct with endogeneous phospholipases (as in Figure 3). The rigidity (saturated lipids, sterol) or other features (high PS-density) of the PM might strongly impede the enzymatic ability of the exogenous construct but likely not the PLCbeta.*

The reviewer's comment hits at a central problem with these experiments: can absence of evidence be evidence of absence? We agree with the reviewer that to make this case, we must

demonstrate that were substrate to be present, we could detect hydrolysis. For the ER and peroxisome, we were unable to conceive of experiments that make this case unambiguously. We therefore did not make definitive conclusions for these organelles with this approach.

For the plasma membrane, we added a new experiment presented in **figure 3B** where we show PM-targeted split PI-PLC has sufficient activity to induce recruitment of the DAG-regulated lipid transfer protein Nir2. These experiments are detailed in our response to reviewer 2's **point 2.5a, on pp. 4-5 of this document.**

As for the reviewer's suggestions to attempt biochemical fractionation, we envision cells could be labelled with 2-[³H]-*myo*-inositol and fractionated, before assaying organelle fractions with recombinant PI-PLC activity. However, we believe the results of these experiments would be difficult to interpret. If enzymatic activity is not observed in a given fraction, it could be because the enzyme is poorly active there, or because the PI is restricted to the luminal leaflet. Furthermore, observation of activity could arise from a pool of PI that flipped during fractionation, or PPI_n that was dephosphorylated (as we discuss in the introduction). Therefore, do not believe it would be possible for such *in vitro* measurements to be usefully correlated with the enzymatic function in intact cells. For these reasons, we did not undertake these specific experiments.

Point 3.2: *A main information is missing in the manuscript and should be provided: the kinetic of recruitment of the targetable split PI-PLC constructs and of other FKBP-tagged lipid-modifying enzymes onto organelles. Is the recruitment of the enzymes is full and immediate after a few seconds or a few minutes? This is necessary to more precisely interpret the kinetics of lipid conversion that theoretically should depend on the level of substrate and amount of enzyme recruited on organelle surface.*

This was a notable omission from the manuscript. We have now added additional **supplementary figures 2, 3 and 4** that detail these kinetics for organelle-targeted PI-PLC^{N187}, FKBP-PI4KA^{C1001} and FKBP-PI4KB, respectively.

We have also detailed these observations for PI-PLC on **p 10, line 25** for mitochondrial-targeted PI-PLC:

Indeed, we could see robust recruitment of PLC^{C100}-FRB-iRFP with a time constant of approximately 3 minutes, which now yielded a substantial increase in C1ab labelling of the mitochondria with similar kinetics (figure S2B, D).

On **p 12, line 8** for PM-targeted PI-PLC:

Dimerization with rapamycin induced translocation of PI-PLC^{C100} to the PM that was largely complete in 1 min (figure S2 B, C) and...

On **p 16, line 1** for PI4KA

Kinetically, recruitment of FKBP-PI4KA^{C1001} occurred with a time constant of 1-2 min to each organelle, with any PI4P biosensor recruitment occurring over a similar or slightly slower time constant (figure S3). In terms of magnitude...

On **p 18, line 2** for the PI4KA at the PM:

In contrast, we could detect no increases in PI4P at the PM by TIRFM after recruiting PI4KA^{C1001} (figure 6A), **despite efficient recruitment within 1 min (figure S3B, C)**, consistent with our observations with PI-PLC (figure 3) and the distribution of TopFluor-PI (figure 1).

On p 18, line 13 for the PI4KB at mitochondria and PM:

We made an FKBP fusion of this enzyme: recruitment to mitochondria **occurred within 1 min and** caused accumulation of PI4P in this membrane with a **time course of approximately 11 min (figure 6B, S4)**, demonstrating the fusion was active. On the other hand, as we observed for FKBP-PI4KA^{C1001}, no increases in PI4P were observed after recruitment of FKBP- to the PM PI4KB (**which occurred within 1 min, figure S4**) relative to an FKBP-only control (figure 6B).

Minor comments

Point 3.3: P3. One sentence is problematic « Therefore, it is currently unclear how much PI is available for PI synthesis in .." »

We thank the reviewer for alerting us to this important typographical error; we have amended this on p 3, line 25:

Therefore, it is currently unclear how much PI is available for **PIP₂** synthesis in cytosolic membrane leaflets

Point 3.4: Figure 1. The name of the markers for the Golgi, ER and mitochondria should be indicated in the panels C, D and E (as in others Figures)

We have made this amendment to **figure 1C-E**.

Point 3.5: Figure 2 and other Figures. The injection of rapamycin should be indicated by an arrow in the plots of panel B and in other Figure showing kinetics of lipid conversion after rapamycin treatment.

We have made this amendment to **figure 2B-C**.

Point 3.6: Figure 2C . Authors should indicate in the Figure to what correspond the green, blue and red channel (iRFP ?) . The cartoon should indicate that the PLCC100-FRBP is tagged with the iRFP

We have made this amendment to **figure 2C**.

Point 3.7: Figure 6 - The recruitment of PI4KA or PI4KB onto the PM does not trigger any production of PI4P or PIP₂, suggesting that no substantial PI pool is present in this membrane. However, considering the initial high level of staining of the PM by the GFP-P4M or mCherry-Tubby constructs (pictures in panel A) , prior to rapamycine addition, is there enough phosphoinositide sensor remaining in the cytosolic to follow any additional synthesis of PI4P or PIP₂ ?

In short, yes. We specifically used low affinity probes for these experiments, notably GFP-P4Mx1 and the low-affinity tubby c-terminal domain mutant, R332H. The images shown in figure 6A show TIRFM images, where the signal relative to the cytosol is not apparent. For both

biosensors, ample cytosolic biosensor remains in the resting state, as documented in the original manuscripts describing these tools (Quinn et al, 2008 and Hammond et al, 2014).

Point 3.8: It is difficult to understand the experiments shown in Figure 6C mainly because the cartoons are a little bit unclear as well as the curves shown on the right (rapamycin is in brown but refer to the blue curve and PI4KB recruitment, the brown curve corresponds to the reference experiment) . Authors must choose between the labeling in brown (rapamycin) or blue (PI4KB). Also, what is the exact order of injection for rapamycin, atropine, carbachol and A1 ?

We have made several modifications to clarify additions common to all conditions (rapamycin, then CCh followed by atropine) with the different conditions (either control, +A1 and +A1 + FKBP-PI4KB):

October 29, 2019

RE: JCB Manuscript #201906127R

Dr. Gerry R Hammond
Department of Cell Biology, University of Pittsburgh School of Medicine
BST-South, Room #327 3500 Terrace St
Pittsburgh, PA 15261

Dear Dr. Hammond,

Thank you for submitting your revised Tools manuscript entitled "Probing the Subcellular Distribution of Phosphatidylinositol Reveals a Surprising Lack at the Plasma Membrane". You will see that the returning reviewers find the revision stronger and recommend publication. We would be happy to publish your paper in JCB pending final revisions necessary to meet our formatting guidelines (see details below). Please also consider the final edits suggested by the reviewers as you prepare your final files.

1) At resubmission, you indicated to us that: "We are also aware that the Balla lab is resubmitting a manuscript on a similar topic; we would be grateful if the journal could delay consideration or a final decision on our paper until such a decision is reached for Balla's in parallel." To ensure confidentiality, we cannot discuss submissions with anyone other than a submission's corresponding author(s). We therefore strongly encourage you to reach out to these authors if you are interested in coordinating publication, and at the submission of your final files, please indicate to us your preference again (whether to hold publication to coordinate it or whether we should proceed). Please feel free to contact us with any questions.

2) eTOC summary: A 40-word summary that describes the context and significance of the findings for a general readership should be included on the title page. The statement should be written in the present tense and refer to the work in the third person.

- Please include a summary statement on the title page of the resubmission. It should start with "First author name(s) et al..." to match our preferred style.

****The eTOC should be revised to match this style**** eg:

Zewe et al develop ways to map the subcellular distribution of phosphatidylinositol (PI) and show that PI is present in most membranes, except for the plasma membrane where it is mainly found as P4P and PI(4,5)P2.

3) Figure formatting: Scale bars must be present on all microscopy images, including inset magnifications. Please add scale bars to 1CDE (magnifications), 2BC (all: main and middle panels), 3A (main and mags), 4A (mags), 5BC (mains and mags), 6B (main and mags), 6C, S1

4) Statistical analysis: Error bars on graphic representations of numerical data must be clearly described in the figure legend. The number of independent data points (n) represented in a graph must be indicated in the legend. Statistical methods should be explained in full in the materials and

methods. For figures presenting pooled data the statistical measure should be defined in the figure legends.

5) Materials and methods: Should be comprehensive and not simply reference a previous publication for details on how an experiment was performed. Please provide full descriptions in the text for readers who may not have access to referenced manuscripts.

- Please include database / vendor IDs for constructs, cell lines, strains (Addgene, ATCC etc.) If no IDs are available, please describe their basic genetic features, even if described in other published work.

- Microscope image acquisition: The following information must be provided about the acquisition and processing of images:

a. Make and model of microscope

b. Type, magnification, and numerical aperture of the objective lenses

c. Temperature

d. imaging medium

e. Fluorochromes

f. Camera make and model

g. Acquisition software

h. Any software used for image processing subsequent to data acquisition. Please include details and types of operations involved (e.g., type of deconvolution, 3D reconstitutions, surface or volume rendering, gamma adjustments, etc.).

6) Please note that tables should be formatted as stand-alone tables, outside of the M&M section. Please include editable files for all tables at resubmission.

7) References: There is no limit to the number of references cited in a manuscript. References should be cited parenthetically in the text by author and year of publication.

- Please abbreviate the names of journals according to PubMed.

8) A summary paragraph of all supplemental material should appear at the end of the Materials and methods section.

- Please include ~1 brief descriptive sentence per item.

9) Conflict of interest statement: JCB requires inclusion of a statement in the acknowledgements regarding competing financial interests. If no competing financial interests exist, please include the following statement: "The authors declare no competing financial interests." If competing interests are declared, please follow your statement of these competing interests with the following statement: "The authors declare no further competing financial interests."

A. MANUSCRIPT ORGANIZATION AND FORMATTING:

Full guidelines are available on our Instructions for Authors page, <http://jcb.rupress.org/submission-guidelines#revised>. **Submission of a paper that does not conform to JCB guidelines will delay the acceptance of your manuscript.**

B. FINAL FILES:

Please upload the following materials to our online submission system. These items are required prior to acceptance. If you have any questions, contact JCB's Managing Editor, Lindsey Hollander

(lhollander@rockefeller.edu).

-- High-resolution figure and video files: See our detailed guidelines for preparing your production-ready images, <http://jcb.rupress.org/fig-vid-guidelines>.

Thank you for this interesting contribution, we look forward to publishing your paper in the Journal of Cell Biology.

Sincerely,

Jodi Nunnari, Ph.D.
Editor-in-Chief, Journal of Cell Biology

Melina Casadio, Ph.D.
Senior Scientific Editor, Journal of Cell Biology

Reviewer #2 (Comments to the Authors (Required)):

The authors have nicely addressed each of the reviewer comments. I have no further comments and congratulate the authors on their work.

Reviewer #3 (Comments to the Authors (Required)):

I consider that the authors responded to most of my comments, notably as they included a number of Supp Figures showing the kinetic of recruitment of the various FKBP-tagged lipid-modifying enzymes to organelle membranes. They are also more careful when commenting the results showed in Fig2. as they now suggest that the soluble split PI-PLC construct and the C1-sensor do

not have necessarily the same catalytic activity and affinity for DAG in the various cell compartments, respectively. Knowing whether lipid sensors recruitment responds in a linear fashion and with the same efficiency, depending on the organelle, to the amount of lipid they detect, remains a difficult question.

As a minor comment, for clarity, authors should modify the Figure S3 to show the kinetics of FKBP-PI4K and its mutated form with colors/line format that are easier to distinguish

I recommend this manuscript for publication